# Kalirin-RAC controls nucleokinetic migration in ADRN-type neuroblastoma

Elena A Afanasyeva[1], Moritz Gartlgruber[1], Tatsiana Ryl[2], Bieke Decaesteker[3], Geertrui Denecker[3], Gregor Mönke[4], Umut H Toprak[1], Andres Florez[1,5], Alica Torkov[1], Daniel Dreidax[1], Carl Herrmann[6], Konstantin Okonechnikov[7], Sara Ek[8], Ashwini Kumar Sharma[9,10], Vitaliya Sagulenko[11], Frank Speleman[3], Kai-Oliver Henrich[1], Frank Westermann[1]

The migrational propensity of neuroblastoma is affected by cell identity, but the mechanisms behind the divergence remain unknown. Using RNAi and time-lapse imaging, we show that ADRN-type NB cells exhibit RAC1- and kalirin-dependent nucleokinetic (NUC) migration that relies on several integral components of neuronal migration. Inhibition of NUC migration by RAC1 and kalirin-GEF1 inhibitors occurs without hampering cell proliferation and ADRN identity. Using three clinically relevant expression dichotomies, we reveal that most of up-regulated mRNAs in RAC1- and kalirin–GEF1–suppressed ADRN-type NB cells are associated with low-risk characteristics. The computational analysis shows that, in a context of overall gene set poverty, the upregulomes in RAC1- and kalirin–GEF1–suppressed ADRN-type cells are a batch of AU-rich element–containing mRNAs, which suggests a link between NUC migration and mRNA stability. Gene set enrichment analysis–based search for vulnerabilities reveals prospective weak points in RAC1- and kalirin–GEF1–suppressed ADRN-type NB cells, including activities of H3K27- and DNA methyltransferases. Altogether, these data support the introduction of NUC inhibitors into cancer treatment research.

## Introduction

Cell migration is the process that occurs during normal embryogenesis, wound healing, immune responses, and metastasis. Accumulated evidence suggests parallelism between metastatic dissemination of tumor cells and migratory processes during embryogenesis. A plethora of genes governing migration during embryogenesis are also involved in the metastatic process (van Zijl et al, 2011). Active cell migration is

essential throughout the whole metastatic process occurring through a sequence of phases including local invasion into the tissue, extravasation into the blood or lymphatic vessels, transit, attachment, intravasation into tissue, colonisation, and proliferation (Tsai & Yang, 2013). The embryonic program of epithelial–mesenchymal transition (EMT) drives cancer cell motility during the dissemination of carcinomas (Thiery et al, 2009). Although transcriptional control and mechanisms of EMT have been elucidated in epithelial cancers, the relevance of EMT to the metastatic process in non-epithelial malignancies, particularly, those of neuroectodermal origin, including neuroblastoma (NB), remains unaddressed. NB, a paediatric malignancy, is thought to be a result of impaired differentiation of neural crest-derived progenitor cells, which promotes the expansion of a population of cells susceptible to the secondary transforming events, that is, deregulation of MYCN via amplification, or c-MYC activation (Westermann et al, 2008; Pei et al, 2013). NBs in children older than 18 mo are metastatic and are associated with a poor survival rate. However, NBs in children younger than 18 mo, especially those with the absence of MYCN amplification and a particular pattern of metastasis (stage 4S), are prone to spontaneous regression and differentiation (Brodeur & Bagatell, 2014). In contrast to other cancers, NB is mostly TP53 wild-type tumor (Chen et al, 2010). NB has been recently resolved as a biphasic malignancy with primary tumors containing the cells of adrenergic (ADRN) type, expressing super-enhancer (SE)–associated transcription factors (TFs) GATA3 and PHOX2B, and mesenchymal (MES) type, expressing SE-associated TFs FOSL2 and RUNX2 (Boeva et al, 2017; van Groningen et al, 2017). MYCN amplification (MNA) correlates with ADRN identity (Gartlgruber et al, 2021). Both ADRN and MES lineages produce aggressive metastatic tumors, whereas ADRN identity could be reprogrammed towards a more chemotherapy resistant MES identity. During mouse development, two neural crest derivatives express Phox2b, Gata3, and other markers of ADRN-type sympathoblasts, and the recently identified

[1]Department of Neuroblastoma Genomics, Hopp-Children's Cancer Center at the (NCT) Nationales Centrum für Tumorerkrankungen Heidelberg (KiTZ), Heidelberg, Germany   [2]Department of Neurosurgery, University of Duisburg Essen, Essen, Germany   [3]Center for Medical Genetics, Ghent University, and Cancer Research Institute Ghent, Ghent, Belgium   [4]European Molecular Biology Laboratories, Heidelberg, Germany   [5]Center for Systems Biology, Faculty of Arts and Sciences, Harvard University, Cambridge, MA, USA   [6]Group of Cancer Regulatory Genomics B086, German Cancer Research Center (DKFZ), Heidelberg, Germany   [7]Department of Pediatric Neurooncology, Hopp-Children's Cancer Center at the (NCT) Nationales Centrum für Tumorerkrankungen Heidelberg (KiTZ), Heidelberg, Germany   [8]Department of Immunotechnology, CREATE Health, Faculty of Engineering, Lund University, Lund, Sweden   [9]Institute for Pharmacy and Molecular Biotechnology and BioQuant, Heidelberg University, Heidelberg, Germany   [10]Division of Theoretical Bioinformatics, German Cancer Research Center (DKFZ), Heidelberg, Germany   [11]School of Chemistry and Molecular Biosciences, The University of Queensland, Brisbane, Australia

Correspondence: elena.afanasyeva@alumni.dkfz.de; f.westermann@dkfz.de

bridge population that connects Schwann cell precursors and mature chromaffin cells (Furlan et al, 2017). Both of these cell types likely give rise to ADRN lineage. In vitro, ADRN type is represented by neuroblastic N- and I-type cells, whereas MES type comprises a group of fibroblast-like, substrate adherent S-cells (Walton et al, 2004; Boeva et al, 2017; van Groningen et al, 2017). Differentiation failure in ADRN NB has been traced at the epigenetic, genetic, and transcriptional level and manifests as the down-regulation of the genes involved in maintaining neuronal morphology (Henrich et al, 2016). Yet, many neuritogenesis genes are essential for cell migration and failure in their regulation might be involved in the NB invasion program. The question remains open as to how differentiation block and migratory propensity are balanced in the ADRN lineage's tumors and whether ADRN-type cells are subject to the EMT process. From a point of clinical relevance, the closest compartment that may reflect migration and dormancy, are disseminated tumor cells (DTCs) detected as part of minimal residual disease (Raimondi et al, 2010; Rifatbegovic et al, 2018). Previous studies in NB identified *DCX* mRNA encoding the core component of neuronal motility, as a robust minimal residual disease marker associated with poor survival in NB patients (Hartomo et al, 2013; Viprey et al, 2014). Based on these facts, we reasoned that understanding the mechanisms implicated in the migration of DCX-positive NB can shed light on the initial steps of the metastatic process in NB. Our data show that *DCX* expression is associated with ADRN identity. Live-cell imaging reveals that migration in ADRN-type cells is coupled with DCX- and LIS1-dependent nucleokinesis (NUC). The silencing or inhibition of RAC1 or ADRN-specific RAC1 guanine nucleotide exchange factor (GEF) KALRN abrogates NUC and, hence, migration. Further analysis with RNAi and chemical compounds reveals kalirin function in coordinating NUC via microtubular (MT) cytoskeleton. These results identify NUC as an important drug target for the development of the migration-specific drug in ADRN-type cells.

# Results

## ADRN NB cells migrate nucleokinetically via DCX- and LIS1-dependent mechanism

We reasoned that, like in neurons, DCX could be part of the migratory program in NB. In primary NB, *DCX* mRNA correlated positively with mRNAs for ADRN SE–associated TFs, *GATA3*, and *PHOX2B*, but not MES *FOSL2*, suggesting ADRN-specific expression ($R_{GATA3}$ = 0.53, $R_{PHOX2B}$ = 0.51; $R_{FOSL2}$ = −0.09; Fig S1A). mRNA profiling in NB cell lines, which enabled detection of ADRN-type and MES-type cells (Fig S1B), confirmed this finding and attributed *DCX* expression to ADRN-type cells (Fig 1A). Chromatin immunoprecipitation sequencing (ChIP-seq) for activation and repression marks demonstrated that *DCX* was transcribed exclusively in ADRN cell lines, whereas the *DCX* locus was silenced in MES cell lines, SH-EP and GI-ME-N (Fig 1B), indicating that ADRN-type cells exclusively express *DCX* mRNA. An inspection of t-SNE–processed expression data from mouse sympathetic precursors (Furlan et al, 2017) showed that *Dcx* was highly expressed by sympathoblasts (Fig S1C). A lower amount of *Dcx* mRNA was present in chromaffin cells and the bridge population. In neuroepithelia (NE) and nascent neurons of the

central nervous system (CNS), DCX specifically regulates nuclear translocations or NUC (Tsai & Gleeson, 2005). We reasoned that NUC could affect the ADRN type of NB cells. Live-cell imaging of MNA ADRN cell lines, IMR-32 and NB-S-124 revealed instances of nuclear mobility (Fig 1C). We pursued this observation further by comparing movies of migrating IMR-32 and SH-EP cells expressing histone H2B fused to a fluorescent protein (nuclear marker or "_NM") and detected variability in nuclear positioning in migrating *TP53*wt (Carr et al, 2006) IMR-32 cells, compared with MES-type SH-EP (Fig 1D and Video 1). Next, we recorded positions of cellular (CC) and nuclear centroids (NCs) (CC; NC tracking). Inspection of these tracks revealed NUC events in IMR-32 cells with the nucleus surpassing the cell centroid, followed by cell contraction or the nucleus leap-frogging over the cellular centroid (Fig 1E). We resolved CC; NC tracks (Lan et al, 2016) by linking each NC from a time point n to the CC in a time point n + 1, generating ∠NCn/NCn+1/CCn+1 (NNC/NCC) angle distribution and NC-CC maps (Fig 1F, left). This analysis revealed an overrepresentation of ≥140°; ≤180° block (140–180°), which should reflect leading process (LP) formation, MES mode and posterior NUC, in SH-EP_NM, as well as the overrepresentation of >0°; ≤40° block (0–40°), which should reflect cell contractions and anterior NUC, in IMR-32_NM and SK-N-BE(2)c_NM (*TP53*mut; Carr et al, 2006) (Fig 1F, left). We decoded NUC events from CC; NC tracks (linkages of the frames: [140–180°; CC > NN], followed by [0–40°; CC < NN]; [0–40°; CC < NN], followed by [0–40°; CC > NN]) which revealed NUC prevalence in IMR-32_NM and SK-N-BE(2)c_NM, compared with SH-EP_NM (Fig 1F, right). Cell velocity in IMR-32_NM and SK-N-BE(2)c_NM cells showed correlation with NUC footprint, compared with SH-EP_NM cells (Fig 1G). To corroborate these findings, we inspected phenotypes of IMR-32, IMR-32_NM, NB-S-124, and three other ADRN cell lines growing on top of three-dimensional collagen (pseudo-3-D assay), and observed traits of neurons migrating through 3-D matrix–bead-like dilations within LPs as well as nuclear deformations (Schaar & McConnell, 2005; Nishimura et al, 2014) (Fig 1H and Video 2). Next, we checked the nuclear migration in IMR-32_NM after cytochalasin B–induced actin depolymerisation or colcemid-induced MT depolymerisation. Both compounds inhibited cell motility; however, cytochalasin B-treated cells retained mobile nuclei, whereas colcemid treatment prevented nuclear migration (Fig S1D and E and Video 3). These data highlighted the role of MT in NUC in ADRN-type cells. Yet, MT-dependent NUC migration is normally observed in postmitotic cells, which leaves open the question as to how this process blends with the cell cycle in NB. In postmitotic neurons, DCX acts jointly with a dynein regulator LIS1 (PAFAH1B1) (Caspi et al, 2000), whereas in proliferating neural progenitors, DCX and LIS1 function differentially during the process named interkinetic nuclear migration (IKNM) (Carabalona et al, 2016). Two correlative studies have already implicated DCX and LIS1 in NB migration (Messi et al, 2008; Evangelisti et al, 2009), which was in support of their concert action in NB. *LIS1* mRNA had a fit with *GATA3*, *PHOX2B*, and *DCX* mRNAs in primary NB ($R_{GATA3}$ = 0.53, $R_{PHOX2B}$ = 0.52; $R_{DCX}$ = 0.42; Fig S1A). Unlike *DCX* mRNA, LIS1 mRNA was present in cancers of non-NE origin and MES-type cells (Fig S1F and G), which could reflect LIS1 function during spindle assembly (Moon et al, 2014). *Lis1* mRNA did not have an affinity with a particular subgroup in t-SNE–resolved expression data from mouse sympathetic precursors (Furlan et al, 2017; Fig S1C). While IKNM was not

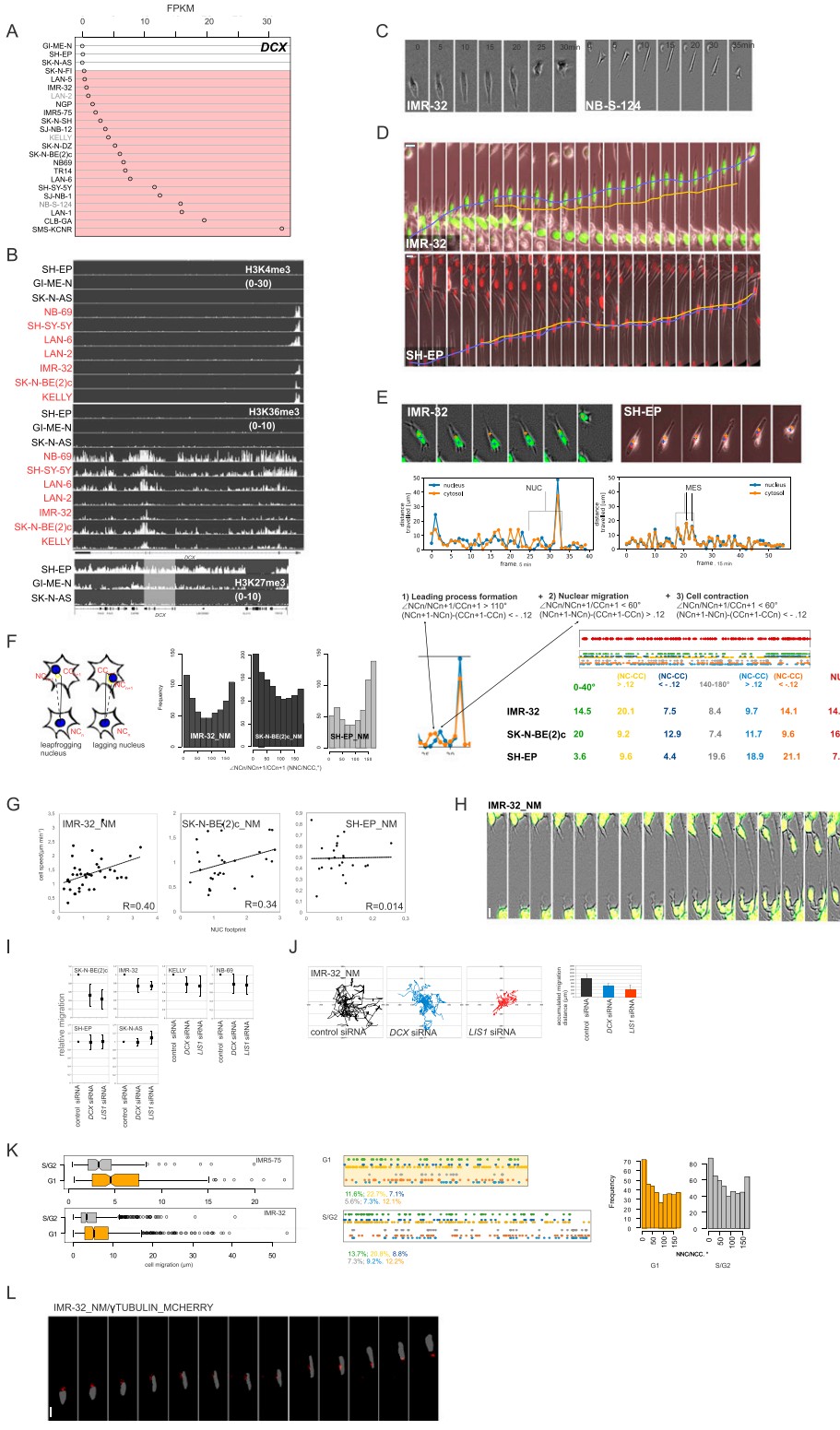

**Figure 1. NB cells exhibit NUC during migration.**
**(A)** *DCX* mRNA expression in MES and ADRN (marked in red) cell lines (the names of ADRN cell lines whose identity was assigned based on the phenotype and/or *PHOX2b*, *GATA3*, and *FOSL2* transcription are represented in grey color [Fig S1A], the cell type identity in the other cell lines [black color] is described in the literature). **(B)** ChIP-seq showing H3K4me3, H3K36me3, and H3K27me3 binding at the *DCX* locus in three MES and ADRN (marked in red) cell lines. For H3K27me3 binding, the region surrounding *DCX* locus (white box) was visualized in IGV program. **(C)** Time-lapse images showing migrating NB-S-124 and IMR-32 cells. **(D)** Time-lapse images of IMR-32_NM and SH-EP_NM cells during migration. Nuclear (in blue) and cellular center (in orange) of migrating cells are indicated. Scale bar 20 µm. **(E)** CN trajectories and CN plots of representative IMR-32_NM and SH-EP_NM cells. **(F)** Schematic of nuclei positioning determined by NCn+1-NCn/|NCn+1-CCn+1 angle (NNC/NCC) (left) and NNC/NCC angle frequency distribution in IMR-32_NM, SK-N-BE(2)c_NM, and SH-EP_NM cells (right). NUC events decoded from CC; nuclear centroid tracks in concatenated tracks from IMR-32_NM, SK-N-BE(2)c_NM, and SH-EP_NM (top right). Mapping of NUC (exemplary track in red), positive and negative noise-corrected NC-CC distances, 0–40° and 130–180° signatures (two or more sequential frames within the same angle block) (exemplary multi-colored track) in concatenated tracks from IMR-32_NM, SK-N-BE(2)c_NM and SH-EP_NM (bottom right). **(G)** Correlation plots between cell velocity and NUC footprint (weighted mean NUC distance) in IMR-32_NM, SK-N-BE(2)c_NM and SH-EP_NM cells. **(H)** Live imaging of pseudo-3-D-assayed IMR-32_NM cells. Scale bar 10 µm. **(I)** 2D exclusion assay in NB cell lines after RNAi against *DCX* or *LIS1* 72 h post-transfection. Relative cell migration is quantified by cell density's normalization to control siRNA-transfected control. Graphs represent the mean relative difference migration ± SD. **(J)** Random walk plots and accumulated migration distances in control IMR-32_NM cells and after RNAi against *DCX* or *LIS1* 72 h post-transfection (13 h, 15-min intervals). Mean migration distances + SD are presented. **(K)** Box plots showing cell migration distances in 216 and 1,528 (G1 phase), 167 and 985 (S/G2 phases) sequential timepoints (20 and 5 min per timepoint, respectively) from tracings of 13 cells from IMR5-75 and 22 cells from IMR-32 expressing the G1 cell cycle sensor (left); *P*-values: IMR5-75: 6.168 × 10⁻⁵, IMR-32: 2.2 × 10⁻¹⁶ (two-sample Kolmogorov-Smirnov test). NNC/NCC angle frequency distribution and noise-corrected nuclear centroid-CC distances (right) in 0–40° and 140–180° signatures in concatenated tracks from IMR-32_NM expressing the G1 cell cycle sensor. **(L)** Time-lapse images showing representative migrating IMR-32_NM expressing γ-tubulin. Scale bar 20 µm.

amenable to examination in dissociated cultures because of the absence of adherens junctions present in vivo (LaMonica et al, 2013), differential knockdown (KD) of *DCX*, controlling G1-specific, kinesin-dependent NUC (Carabalona et al, 2016), and of *LIS1*, controlling G2-specific, dynein-dependent NUC, and spindle assembly during IKNM (Tsai et al, 2005; Yingling et al, 2008; Carabalona et al, 2016), could help to tell IKNM from NUC in NB. 2D exclusion assays revealed inhibition of migration after either *DCX*

or *LIS1* KD in ADRN cell lines (Fig 1I). Proliferation in NB cell lines was not affected in *DCX-KD* and *LIS1*-KD cells (Fig S1H), which ruled out DCX and LIS1 involvement in spindle assembly or cell cycle transition in NB. Our tracking of randomly walking cells in *DCX-KD* and *LIS1*-KD IMR-32_NM spheroids showed inhibition of motility (Fig 1J and Video 4; data not shown). We then tested cell cycle specificity of migration in ADRN-type cells by performing cell tracking in asynchronously growing IMR-32 and an IMR-32 derivate, IMR5-75 expressing a FUCCI cell cycle sensor (Ryl et al, 2017), which revealed a tendency for migration in G1 phase (Fig 1K, left). No significant difference in NNC/NCC angle distributions in G1 and S/G2 phases was observed in IMR-32_NM (Fig 1K, right). Expression of a construct encoding γ-tubulin fused to mCherry in IMR-32 showed that nuclei surpassed γ-tubulin signals during migration (Fig 1L and Video 5), which was in agreement with the nucleus-centrosome (N-C) inversion mechanism (Umeshima et al, 2007).

The plotting of nuclear positions (NCn-CCn or |NC-CC|) showed that nuclei were less present in the periphery of the cells after *DCX*- or *LIS1*-KD (Fig 2A). Staining for NUC-relevant neuronal βIII-tubulin (Xie et al, 2003) and γ-tubulin showed that cell phenotypes produced by RNAi of *DCX* and *LIS1* were different and resembled neurons after *Lis1* or *Dcx*-KD (Youn et al, 2009; Nishimura et al, 2014), that is, higher variability in the N-C distance and dose-dependent unipolar neurite outgrowth in *LIS1*-KD cells as well as defective LPs in *DCX*-KD cells (Fig 2B). Accordingly, suppression of 0–40° signature, NUC as well as no NNC/NCC angle overrepresentations were seen after *DCX-KD* and *LIS1*-KD, respectively (Fig 2C and D). Yet, cell velocity in *DCX-KD* and *LIS1*-KD IMR-32_NM cells showed substantial correlation with the NUC footprint (Fig 2E). Altogether, these analyses showed no evidence for MES migration mode in *LIS1-KD* and *DCX-KD* IMR-32_NM cells. Also, gene set enrichment analysis (GSEA) of *DCX*- and *LIS1*-KD IMR-32 RNA-seq-resolved expression profiles showed that there was no induction of the MES program (Fig S2A and Table S1), which supported our idea regarding the maintenance of ADRN program after *DCX*- and *LIS1* RNAi. *DCX*-KD and *LIS1*-KD transcriptomic profiles showed remarkable overlap (Fig 2F). Particularly, MT-related gene signatures and gene signatures associated with cortical cytoarchitecture were depleted in *DCX*-KD and *LIS1*-KD transcriptomes (Table S1). Gene signatures related to nonsense–mediated decay (NMD) and mRNA transport, mitochondrial function, oxidative phosphorylation (oxphos) signature, as well as genes related to cell contraction and Ras signalling regulation were specifically depleted in *DCX*-KD and *LIS1*-KD cells, respectively (Figs 2G and S2B). The down-regulation of pathways related to mitochondrial function was the only expression trait shared with KD of a DCL family gene, *dclk1* in mouse neuroblastoma (Verissimo et al, 2010) (Table S1). On the other hand, GSEA of the up-regulated genes showed less consistency even under the relaxed threshold (q-values ≤ 0.25) (Table S1). Pseudogene transcripts, signatures related to G-protein coupled receptor signalling and PTPRB neighbourhood were overrepresented by the up-regulated genes in *DCX*-KD cells (Fig S2C and Table S1). Next, we assessed *DCX*- and *LIS1*-KD expression profiles in the context of primary NB using three clinically relevant dichotomies (stage 4S versus stage 4, stages 1|2 versus stage 4; *MYCN*-nonamplified versus MNA tumors). This analysis showed that up-regulated, but not down-regulated genes in the *DCX*-KD cells had an affinity toward transcriptomic profiles of stage

4S, stages 1|2 and *MYCN*-nonamplified tumors (Figs 2H and S2D). We then extracted then the genes from *DCX*- and *LIS1*-KD profiles that did not recapitulate association with low-risk characteristics, which we named "mis-expressed" genes (*DCX(LIS1)*-KD$^{UP}$ ∩ stage 4S [stages 1|2; *MYCN*-nonamplified]$^{DOWN}$; *DCX(LIS1)*-KD$^{DOWN}$ ∩ stage 4S [stages 1|2; *MYCN*-nonamplified]$^{UP}$ [*P*-values ≤ 0.05, no logFC threshold]). GSEA showed that mis–down-regulation engaged TP53 targets and mis–up-regulation engaged CHEK2 neighbourhood in *LIS1*-KD cells (Fig 2I and Table S1). Given the association of *LIS1* expression with favourable prognosis in NB (Garcia et al, 2012), these results might reflect LIS1 involvement in the potentially tumor-suppressive process(es) in NB. In *DCX*-KD cells, little consistency was found in the list of mis–up-regulated genes, whereas mis–down-regulation engaged genes encoding extracellular matrix, MYC targets as well as genes marked bivalently in the CNS (Fig 2J). These observations showed that phenotypic NUC inhibition in ADRN-type NB could generate disparate transcriptomic alterations. In *DCX*-KD cells, the down-regulated genes showed positional enrichment for NB-relevant regions of loss of heterozygosity (Mora et al, 2001; White et al, 2001; Lasorsa et al, 2020), including 19p13, 19q13, and 1p36 (Table S1). We checked the literature on NB for similar observations and found that a link between *MYCN* down-regulation and oxphos inhibition was identified previously in MNA NB cells (Dzieran et al, 2018; Oliynyk et al, 2019). Also, the positional depletion of 19p13 genes was observed in disseminated NB tumor cells (NB DTC) from relapsed patients (Table S8 by Rifatbegovic et al, 2018). Although the 19p13 gene overlap in the *DCX*-KD and NB DTC profiles was negligible (Table S1), 19p13 shutdown in relapse DTCs and *DCX*-KD cells could imply similar regulatory mechanisms. We retrieved the RNA-seq data from NB DTCs and reanalysed with the following settings: no logFC cutoff; *P*-values ≤ 0.05; DTC$^{TOTAL}$ versus Tumor (TU)$^{TOTAL}$; DTC$^{MNA}$ versus TU$^{MNA}$; DTC$^{RELAPSE}$ versus TU$^{RELAPSE}$ to identify whether similarities between transcriptomic profiles of *DCX*-KD IMR-32 and NB DTC go beyond the 19p13 genes. *DCX*-KD$^{UP}$ gene set showed splitting when mapped onto NB DTC transcriptome profiles, whereas *DCX*-KD$^{DOWN}$ showed a match with NB_DTC$^{DOWN}$ (Fig 2K). Cross-checking for immune cell-specific signatures showed that the presence of *DCX*-KD$^{UP}$ ∩ NB_DTC$^{UP}$ overlap could not be explained by bone marrow cell contamination (Fig S2E). Several gene signatures associated with stage 4S, stages 1|2 and *MYCN*-nonamplified status, as identified by the parametric analysis of gene set enrichment (PAGE analysis) using gse49710 signature (Table S1), showed similar splitting (Fig S2F), suggesting cellular heterogeneity of NB DTCs rather than a gene signature functional variegation.

Altogether, the results show that migration in ADRN-type cells depends on the N-C inversion mode of NUC that requires both DCX and LIS1, which rules out the possibility of proliferative NUC, IKNM. The results also imply that *DCX*-KD–like situations appear in disseminated NB cells.

## ADRN SE-associated TF, SOX11, is involved in NUC regulation in ADRN NB

The parallelism between migration modes in neurons and ADRN-type NB cells is likely to extend to the transcription control, which could involve neuron-specific TFs embedded in NB pathogenesis. We reasoned that DCX co-expression signature may shed information on NUC control in NB, as the expression of *DCX* mRNA was a specific trait of cancers of NE origin, including NB (Figs 3A, top and

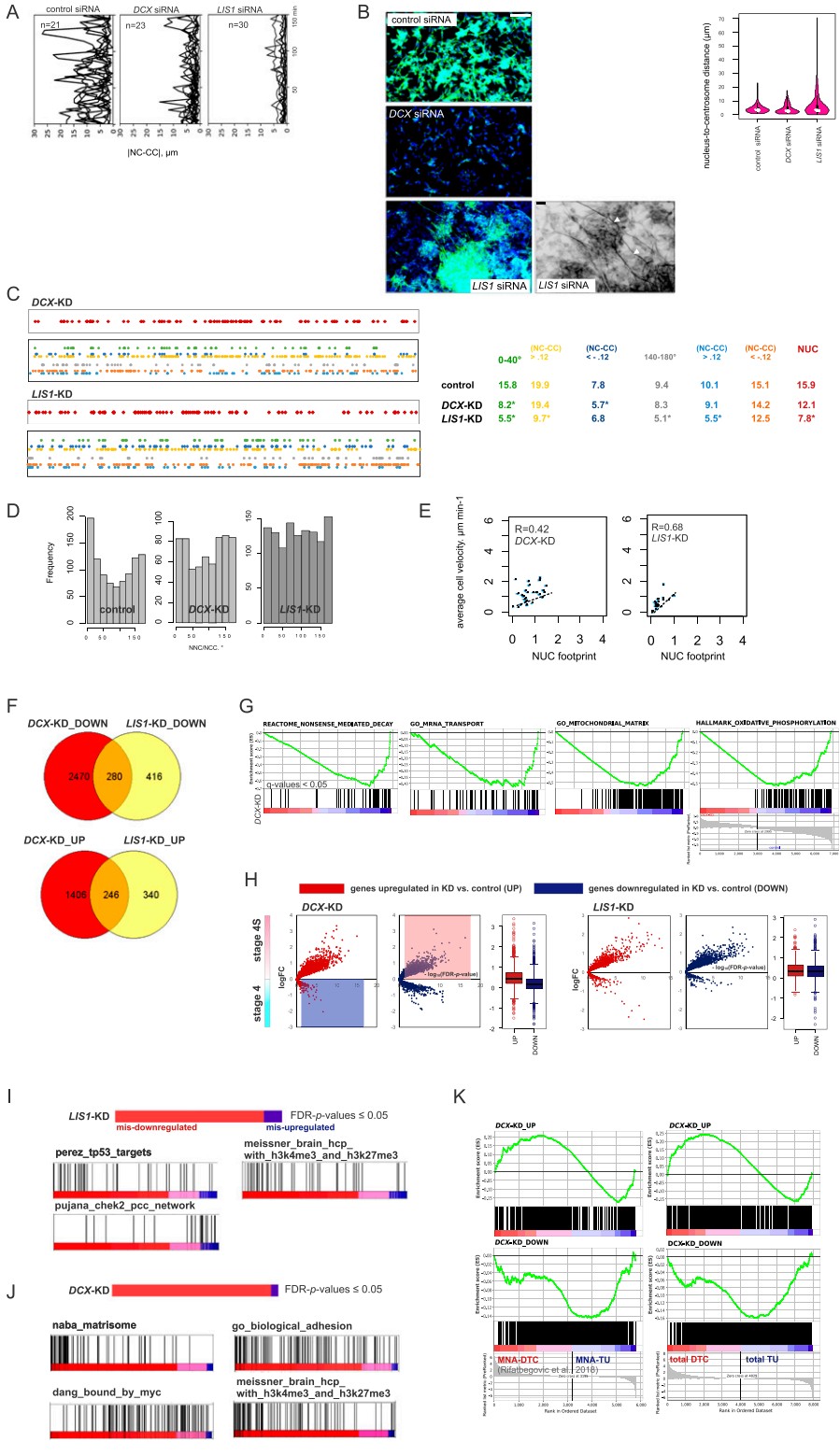

**Figure 2. NUC genes are involved in migration in ADRN-type cells.**
**(A)** |NC-CC| plots for control IMR-32_NM cells and the cells after *DCX*- or *LIS1-KD* 72 h post-transfection. **(B)** βIII-tubulin immunolabeling (top left) in control IMR-32 cells and after RNAi against *DCX* and *LIS1* 72 h post-transfection. Neurite outgrowth after *LIS1* RNAi (bottom left; grayscale negative field) is indicated by white arrows. Scale bar 100 μm. Violin plots showing the nucleus-to-centrosome distance distribution in *DCX*- and *LIS1-KD* IMR-32 cells (right). Means are indicated. *P*-values: *DCX* siRNA: 0.009898, *LIS1* siRNA: n.s.; (Welch *t* test). **(C)** Mapping of NUC, positive and negative noise-corrected NC-CC distances, 0–40° and 140–180° signatures in concatenated tracks from *DCX-KD* (27 cells, 762 time points) and *LIS1-KD* (41 cells, 1,256 timepoints) IMR-32_NM. **(D)** NNC/NCC angle frequency distribution in concatenated tracks in *DCX*- and *LIS1-KD* IMR-32_NM cells. **(E)** Correlation plots between cell velocity and NUC footprint in *DCX*- and *LIS1-KD* IMR-32_NM cells. **(F)** Venn diagram showing overlaps between the differentially expressed genes (DEGs; |logFC| cutoff: 0.3) in *DCX*- and *LIS1-KD* IMR-32 versus control IMR-32. **(G)** Gene set enrichment analysis plots of the indicated gene sets in *DCX*-KD IMR-32. False discovery rate (FDR)-adjusted *P*-values (q-values) are listed. **(H)** Volcano plots showing log2FC expression (y-axis) and log10 FDR-adjusted *P*-value (−log10 FDR-*P*-value, x-axis) of DEGs in *DCX*- and *LIS1*-KD IMR-32 in stage 4S versus stage 4 tumors (GEO: gse49710). Each dot represents an individual spot. **(I, J)** Gene set enrichment analysis plots of the indicated gene sets in *LIS1*- and *DCX*-KD IMR-32 based on their mis-expression in stage 4S versus stage 4 tumors (*P*-values by two-way *t* test ≤ 0.05, no logFC cutoff). **(K)** Plots showing mapping of DEGs of *DCX*-KD IMR-32 onto NB disseminated tumor cell transcriptome profiles (Rifatbegovic et al, 2018).

S1A, and S3A). *DCX* expression did not discriminate between low-stage and advanced-stage NB tumors (Fig S3B), which could emerge from differential TF control; therefore, we focused on the stage 4 subset (gse49710). Interrogation of *DCX* co-expression signature for the general TFs involved in neuronal migration (Kwan et al, 2012; Hoshiba et al, 2016) retrieved five prospective candidates (Fig S3C). One candidate, the high-mobility-group domain-containing TF encoding *SOX11* was a strong hit (ninth most positively *DCX*

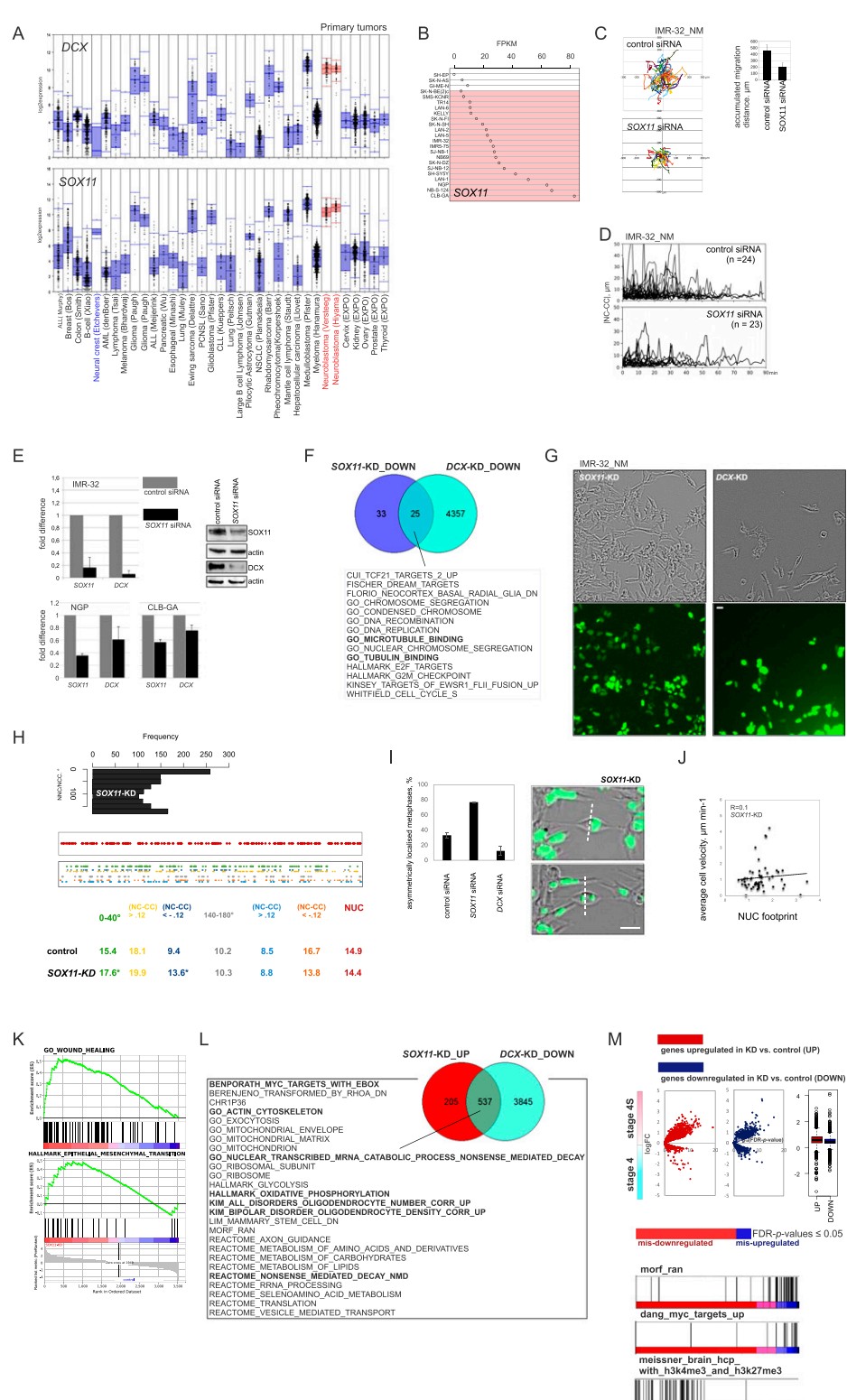

**Figure 3. SOX11 regulates NUC.**
**(A)** Affymetrix *DCX* mRNA (probe set 229349, top) and *SOX11* mRNA (probe set 204914, bottom) expression analysis from primary tumors. **(B)** *SOX11* mRNA expression in MES and ADRN (marked in red) cell lines. **(C)** Random walk plots and accumulated migration distances in control IMR-32_NM and after RNAi against *SOX11* 72 h post-transfection (6 h, 15-min intervals). Mean migration distances + SD are presented. The efficiency of *SOX11*-KD was determined by WB. **(D)** Representative |NC-CC| plots in IMR-32_NM controls and after RNAi against *SOX11*. **(E)** Relative qRT-PCR for *SOX11* and *DCX* in IMR-32 after *SOX11* RNAi 72 h post-transfection and in NGP and CLB-GA after *SOX11* RNAi 48 h post-transfection (left). Mean relative difference values + SE of control are presented. WB for DCX in IMR-32 after 48 h post-transfection (right). **(F)** Venn diagram showing the number of gene sets (extracted from MSigDB [22,596 gene sets]) in IMR-32 *DCX*-KD[DOWN] ∩ *SOX11*-KD[DOWN] semantic overlap (q-values ≤ 0.05). **(G)** Cell morphology in *SOX11*-and *DCX*-KD cells. Scale bar 20 μm. **(H)** NNC/NCC angle frequency distribution (top) and mapping of NUC, positive and negative noise-corrected NC-CC distances, 0–40° and 140–180° signatures (bottom) in *SOX11*-KD (57 cells, 1,314 timepoints). **(I)** The percentages of asymmetrically localised metaphases (left) in control, *SOX11*- and *DCX*-KD IMR-32_NM. Mean percentages + SD are presented. Representative images of dividing *SOX11*-KD cell (right, scale bar 20 μm). **(J)** Correlation plot between cell velocity and NUC footprint (weighted mean NUC distance) in *SOX11*-KD IMR-32_NM. **(K)** Gene set enrichment analysis plots showing gene sets "wound healing" and "epithelial–mesenchymal transition" in *SOX11*-KD versus control. **(L)** Venn diagram showing the number of gene sets (MSigDB) in IMR-32 *SOX11*-KD[UP] ∩ *DCX*-KD[DOWN] semantic overlap (q-values ≤ 0.05). **(M)** Volcano plots showing expression of DEGs in *SOX11*-KD IMR-32 in stage 4S versus stage 4 tumors.

mRNA-correlated TF in stage 4; $R_{stage4}$ = 0.54). *SOX11* is mutated in Coffin-Siris syndrome (Tsurusaki et al, 2014; Nemani et al, 2014) and is a likely candidate for 2p25.2 deletion syndrome (Lo-Castro et al, 2009); these disorders are characterised by

microcephaly with hindbrain abnormalities. In mice, Sox11 deficiency generates complex CNS defects that stem from proliferation deficits in NE and neuronal migration errors (Hoshiba et al, 2016). In peripheral nervous system (PNS), Sox11 regulates proliferation during

the early development of sympathetic ganglia (Potzner et al, 2010). It was previously shown that Sox11 was indispensable for the proliferation of tyrosine hydroxylase-expressing precursors in developing sympathetic ganglia (Potzner et al, 2010). In line with this, *Sox11* expression was observed in sympathoblasts and the bridge population (Furlan et al, 2017; Fig S3D). In NB, *SOX11*, identified previously as an SE-associated TF in ADRN-type cells (van Groningen et al, 2017), was highly expressed in *MYCN*-nonamplified stage 4 and MNA NB (localised [LOC] versus *MYCN*-nonamplified stage 4, *P*-value < 0.005; *MYCN*-nonamplified versus MNA, *P*-value < 0.001; one-way ANOVA test; Fig S3E). *SOX11* expression profile across cancer cell lines and primary tumors showed concordance with *DCX* expression, except for human neural crest cells and pheochromocytoma, which could partially reflect earlier *SOX11* induction during neural crest development and SOX11 lineage specificity (Potzner et al, 2010; Figs 3A, bottom and S3F). Consistently, *SOX11* mRNA was prevalent in ADRN cell lines, as compared with MES cell lines (Fig 3B). SOX11 directly controls *DCX* expression in neurons (Mu et al, 2012), which could also hold true in NB and, hence, affect cell migration. We observed decreased migration from *SOX11*-KD IMR-32_NM spheroids and the closure of |NC-CC| distance along with *DCX* downmodulation (Fig 3C–E and Video 4). Real-time qRT-PCR in two other *SOX11* high expressors, ADRN cell lines CLB-GA and NGP, also showed *DCX* mRNA depletion after *SOX11* RNAi (Fig 3E). As RNA-seq data showed, SOX11 targets identified in mouse ES-derived nascent neurons (Bergsland et al, 2011) were significantly down-regulated (q-value < 0.05), whereas other SOX11 target datasets (Lachmann et al, 2010; Kuo et al, 2015) did not demonstrate depletion (Fig S3G). *DCX*- and *SOX11*-KD transcriptomic profiles showed little overlap, but formed a *DCX*-KD^DOWN ∩ *SOX11*-KD^DOWN semantic match that involved signatures related to MT function and mRNA transport as well as MYC targets (Fig 3F). No overlapping signatures were found in the up-regulated genes (Table S2). In contrast to the findings in neurons (Piens et al, 2010; Mu et al, 2012), *DCX* promoter was not bound by SOX11 in NB cells (Decaesteker et al, 2020 Preprint). Also, *SOX11*-KD IMR32_NM had different morphology, as compared with the *DCX*-KD cells (Fig 3G) and retained parental NNC/NCC angle distribution (Fig 3H, top). The impact of 0–40° signature and 0–40°-linked NUC was preserved in *SOX11*-KD cells (Fig 3H, bottom). Given the intact NNC/NCC angle distribution, it was worth checking nuclei localisation in the pre-mitotic KD cells, which revealed a higher percentage of asymmetrically localised metaphases in *SOX11*-KD IMR-32_NM, compared with *DCX*-KD and control cells (Fig 3I). Cell velocity in *SOX11*-KD IMR-32_NM cells showed no correlation with NUC footprint, compared with control or NUC-suppressed, *DCX*-KD and *LIS1*-KD cells (Figs 3J and 2E). This suggested that *SOX11*-KD IMR-32_NM acquired slower, NUC-independent migration mode. In line with this, we observed the overrepresentation of gene hallmarks "wound healing" and "EMT" by the up-regulated genes, which supported our idea about *SOX11* RNAi-induced reprogramming (Fig 3K and Table S2). *SOX11* RNAi resulted in down-regulation of several ADRN TFs, *ISL1*, *KLF7*, *MYCN*, and neuron-specific RNA binding protein-encoding *ELAVL2* and *ELAVL4*, as well as induction of MES TFs, *ETS1* and *JUND* (Boeva et al, 2017; van Groningen et al, 2017; Zeid et al, 2018) (Table S2) and the targets of ETS-1 and AP-1. The targets of ELAVL proteins identified previously in IMR-32 (Scheckel et al, 2016) were down-regulated in *SOX11*-KD IMR-32, which was in striking contrast to

the *DCX*-KD profile (Fig S3H). Also, gene signatures "oxphos," "NMD" and gene lists associated with migration/actin cytoskeleton regulation and cortical cytoarchitecture showed inverted pattern of regulation in *SOX11*-versus *DCX*-KD cells (*SOX11*-KD^UP ∩ *DCX*-KD^DOWN) (Fig 3L and Table S2). Further assessment of *SOX11*-KD transcriptomic profile in the context of the clinically relevant dichotomies revealed that mis–up-regulation engaged MYC targets, RAN neighbourhood, whereas mis–down-regulated involved bivalently marked genes (Figs 3M and S3I and Table S2). The *SOX11*-KD^DOWN gene set matched with NB DTC^UP set (Fig S3J). This indicated that SOX11^HIGH, rather than SOX11^LOW IMR-32 reflected a NB DTC subcompartment. Next, we inspected whether SOX11 was capable of initiating the MES-to-NUC transition by forcibly expressing SOX11 in the inducible format in SH-EP. We observed a moderate velocity gain in the cells with enforced SOX11 expression, compared with the controls (Fig S3K). NNC/NCC angle distribution revealed acquisition of 0–40° block in SH-EP expressing SOX11, but the highest cell velocity was associated with 140–180° block in SH-EP expressing SOX11 (Fig S3L), suggesting that SOX11 forced expression interfered with nuclear positioning mechanisms in SH-EP without reprogramming migration type.

Altogether, these data show that NUC is controlled by SOX11 in NB. *SOX11* RNAi fosters morphological asymmetric cell divisions and causes reprogramming of NUC migration in ADRN-type NB cells. The latter was in contrast to the observations carried out in nascent neurons previously.

## Inhibition of ROCK or RAC1 blocks different steps of NUC migration

We reasoned that targeting the intrinsic regulators of the MT cytoskeleton to generate *DCX* RNAi-like NUC errors was a prospective direction for the development of migration blockers in the ADRN-type NB. In neurons, the formation of LPs is regulated by RAC1, whereas actomyosin contractions at the trailing end are regulated by RHOA-ROCK (Kawauchi et al, 2003; Martini & Valdeolmillos, 2010). The down-regulation of RAC1 neighbourhood, RHOA-related gene signatures, "BERENJENO_transformed_by_RHOA_DN," and "BERENJENO_transformed_by_RHOA_UP," after *DCX* RNAi indirectly supported the role of the RAC1 and RHOA pathways in *DCX*-KD phenotype in NB (Fig 4A). To assess these processes, we applied a ROCK inhibitor, Y27632 (5 μM), which reduced migration in all tested ADRN and MES cells, whereas cell viability was unaffected (Figs 4B, left and S4A and B). In contrast, an RAC1 inhibitor, NSC23766 (10 μM) (Bid et al, 2013), reduced migration only in ADRN cell lines without affecting cell viability (Figs 4B, left and S4B and C), in line with the observed effect upon RAC1 siRNA treatment (Figs 4B, right and S4D). In neurons, the RAC1 function is mainly associated with the modulation of cytoskeletal dynamics in the growth cones, dendritic spines, and lamellipodia, which are absent in IMR-32 (Fig S4E, left). In line with this, per equal amount of total protein, basal RAC1 activity was lower in ADRN IMR-32 and SK-N-BE2c, as compared with SH-EP (Fig S4E, right). The absence of lamellipodia was reflected by perinuclear and nuclear localisation of RAC1 identified with either immunostaining in IMR-32 or forced expression of fluorescent protein-tagged full-length RAC1 (Fig 4C). Visually, only ROCK inhibitor-exposed ADRN-type cells underwent a

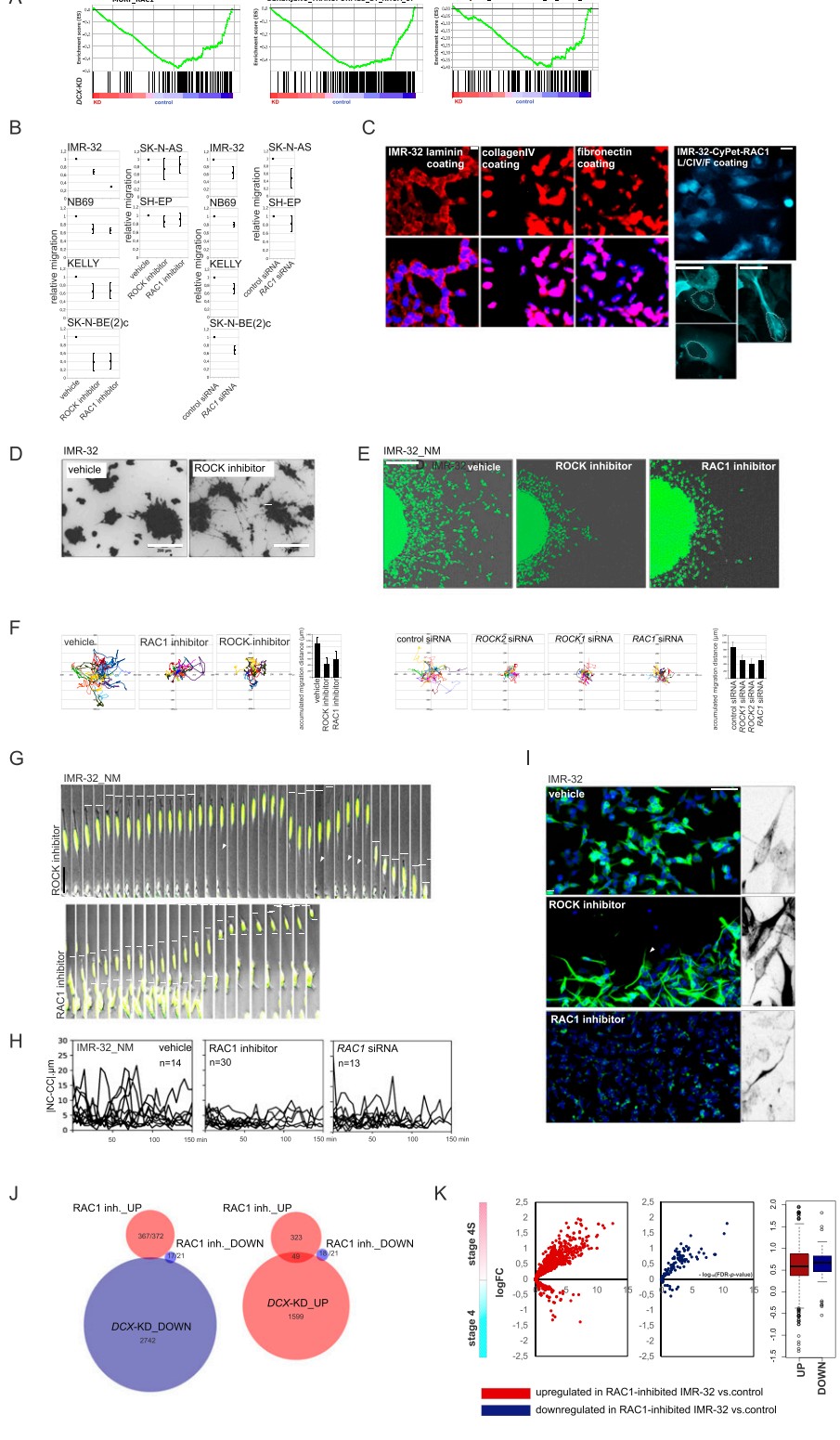

**Figure 4. ROCK and RAC1 inhibition interferes with cell detachment and NUC.**
**(A)** Gene set enrichment analysis plots showing RHOA-related and RAC1-related gene sets in *DCX*-KD IMR-32 versus control. **(B)** 2D exclusion assay in NB cell lines after treatment with the RAC1 inhibitor or ROCK inhibitor and after *RAC1* RNAi (left). Relative cell migration (right) is quantified by normalization of cell density to vehicle- or control siRNA-treated cells. Mean relative difference values ± SD are presented. **(C)** RAC1 immunostaining and CyPet-RAC1 subcellular localisation in IMR-32 (L, laminin; CIV, collagen IV; F, fibronectin). Scale bar 20 *µm*. **(D)** Images of IMR-32 spheroids stained with Calcein AM after 72 h of treatment with vehicle or ROCK inhibitor in pseudo 3-D. **(E)** Random walk plots in IMR-32_NM treated with vehicle, RAC1 inhibitor or ROCK inhibitor for 72 h (13 h, 15-min intervals). Scale bar 200 *µm*. **(F)** Accumulated migration distances of control, ROCK inhibitor– and RAC1 inhibitor–treated IMR-32_NM cells and after *ROCK1*, *ROCK2*, or *RAC1* RNAi. Mean migration distances + SD are presented. **(G)** Time-lapse images of IMR-32_NM after treatment with ROCK inhibitor (left) or RAC1 inhibitor (right). Nuclei and cell leading edges of migrating cells are indicated. Scale bar 20 *µm*. **(H)** |NC-CC| plots in control IMR-32_NM and after treatment with the RAC1 inhibitor or RNAi against *RAC1*. **(I)** *β*III-tubulin immunolabeling in IMR-32 after treatment with vehicle, ROCK- or RAC1 inhibitor. Fork-like structures are showed in grayscale negative field. Scale bar 100 *µm*. **(J)** Venn diagrams showing the numbers of genes in *DCX*-KD ∩ RAC1 inhibitor transcriptomic overlap (*P*-values ≤ 0.09; |logFC| cutoff [*DCX*-KD]: 0.3, |logFC| cutoff [RAC1 inhibitor]: 0.2). **(K)** Volcano plots showing expression of DEGs in RAC1-inhibitor-treated IMR-32 in stage 4S versus 4 tumors.

profound morphological transformation (Figs 4D and S4F). Tracking showed motility defects after either RAC1 or ROCK inhibition in IMR-32_NM, which was further confirmed by RNAi against *ROCK1*, *ROCK2*, and *RAC1* (Figs 4E and F, and S4G and Video 5). Both RAC1- and ROCK inhibition–induced defects in IMR-32_NM fit the neuroblastic cell phenotype and did not demonstrate MES traits (Fig 4G and Videos 5–Videos 7). Cell morphology after *RAC1* suppression revealed NUC defects, which was also confirmed by the results of |NC-CC| plotting

(Fig 4H). In agreement with RAC1/ROCK antagonism (Petrie et al, 2009), RAC1 inhibitor-treated cells maintained cell contractions, whereas ROCK inhibitor–exposed cells retained nuclear migrations but failed to detach (Fig S4H and Videos 5–Videos 7). Staining for βIII-tubulin revealed that ROCK inhibition up-regulated NUC fork–like MT elements (Xie et al, 2003), whereas fewer MT structures were observed after RAC1 inhibition, which was an indication of RAC1 involvement in the regulation of NUC MTs (Fig 4I). RNA-seq analysis showed that there was an overlap between transcriptomic alterations induced by RAC1 inhibitor and DCX-KD (Fig 4J). Similar to DCX RNAi, genes up-regulated by RAC1 inhibition genes had an affinity towards stage 4S, stages 1|2 and MYCN–nonamplified status (Figs 4K and S4I). GSEA showed that the genes up-regulated by RAC1 inhibition were enriched for TCF21-dependent genes and bivalently marked genes (Table S3). Although we did not find GSEA-based evidence for MES signature in the transcriptomic profile of RAC1-inhibited cells, several hallmarks, including ETS fusion (EWS/ETS) targets, formed an overlap with the SOX11-KD$^{UP}$ gene set. Also, RAC1 neighbourhood genes that showed inverted, DCX-KD$^{DOWN}$/SOX11-KD$^{UP}$, expression pattern, were not depleted in the RAC1 inhibitor-treated cells (Table S3). We thought that identification of NUC RAC1-GEF(s) could help to refine RAC1 suppression, and hence, inhibition of NUC migration in the context of NB. As previous studies showed, three GEFs, T-lymphoma and metastasis gene 1 TIAM1, triple functional domain protein TRIO and TRIO homologue kalirin, are inhibited by NSC23766 (Gao et al, 2004; Zeinieh et al, 2015). These GEFs, which were already implicated in NB biology (Molenaar et al, 2012; Pugh et al, 2013), were highly expressed in primary NB and NB cell lines (Fig S5A and B). To date, only Tiam1 was reported to activate Rac1 in the context of migrating mouse CNS neurons (Kawauchi et al, 2003). In primary NB, KALRN showed stronger co-expression with DCX and SOX11, compared with TRIO and TIAM1 (Fig S5C), indicating ADRN-type–specific expression of KALRN, which was also reflected by RNA profiles in NB cell lines (Fig 5A). We checked the expression of the three GEFs by WB using RAC1–GEF domain-relevant ABs, which showed a 160-kD band, identified as delta-kalirin-8 (Mains et al, 1999, in primary MNA NB as well as ADRN cell lines, and a 115-kD band in several MYCN-nonamplified primary NBs (Figs 5B and S5D). An AB against kalirin PDZ-binding motif (STYV) detected a 100-kD delta-kalirin-7 in several primary MYCN-non-amplified NB. TIAM1 isoforms were found in ADRN cell lines as well as several primary NBs. An anti-TRIO AB detected several annotated isoforms in one primary MYCN-amplified NB and a 50-kD band in the cell lines. As immunostaining showed, kalirin isoforms, like RAC1, had perinuclear and nucleolar localisation, whereas TRIO and TIAM1 were found only in nucleoli and in cell boundaries, respectively, which supported the likelihood of kalirin involvement in RAC1 activation and MT regulation (Figs 5C and S6A–C). TIAM1, TRIO, and KALRN are potentially regulatable by SOX11 (Table S2); therefore, we checked their expression by WB, which revealed a reduction of kalirin and TIAM1 levels in SOX11-KD IMR-32 cells (Fig S6D). Kalirin-SP and kalirin-STYV signals had uneven distribution at migration fronts of immunostained NB-S-124 (Fig S6E) and IMR-32 spheroids (data not shown), which was also confirmed by WB of sparsely seeded cells, suggesting that kalirin isoforms were functionally diversified (Fig 5D). We down-regulated the kalirin RAC1-GEF (GEF1) pharmacologically with the compounds that do not interfere with TIAM1–ITX3 (kalirin–GEF1 inhibitor#1, 10 μM) and NPPD (kalirin–GEF1 inhibitor#2, 5 μM) (Blangy et al, 2006; Ferraro et al, 2007; Bouquier et al, 2009; Yan et al, 2015). Active RAC1 levels were lower in IMR-32 and SK-N-BE(2)c treated with either RAC1 inhibitor or each of kalirin–GEF1 inhibitors. KALRN RNAi in IMR-32 (Fig S7A and B) also reduced RAC1 activity, thus confirming kalirin involvement in RAC1 activation (Figs 5E and S7C). This effect was not observed in the reprogrammed after SOX11 RNAi cells (Johnson et al, 2000).

Based on co-expression with DCX mRNA and WB results, several ADRN type-specific KALRN transcripts were produced in primary NB, which was confirmed by ChIP-seq profiling for H3K36me3 in NB cell lines (Fig 5F). H3K4me3 peaks' presence at exon-B and RAC1–GEF–unrelated Duet exon in all tested cell lines along with H3K27me3 loading at the KALRN in MES cell lines was indicative of bivalency, associated with the genes involved in neuronal specification (Liu et al, 2017). Kalirin is expressed in the murine heart, adrenal medulla and superior cervical ganglia (May et al, 2002). Kalrn mRNAs demonstrated no affinity with a particular population on the t-SNE–processed expression data from mouse sympathetic precursors (Furlan et al, 2017; Fig S7D). More specifically, kalirin isoforms –9 and –12 are expressed in mature sympathetic neurons (May et al, 2002), kalirin-9 protein is expressed in cardiac outflow tract (Wu et al, 2013), whereas kalirin-8 protein is found in rat and mouse neuroendocrine cells (Hansel et al, 2001; Ferraro et al, 2007). ADRN-type cells are likely to inherit KALRN expression from a sympathoadrenal precursor. In line with this, kalirin-9 and -12 expression was higher in stage 4S NB (4S versus 4: P-value = 4.4 × 10$^{-6}$ [kalirin-9]; P-value = 9.1 × 10$^{-3}$ [kalirin-12]) (Fig 5G). Given the low contribution of Duet 5′-exons into KALRN isoform repertoire in NB, this indicated that kalirin-9 was the most highly expressed KALRN isoform in stage 4S. 3′-UTR diversity was identified for kalirin-9 because a sub-isoform with a cryptic 3′-UTR exon carrying a stop codon was present in IMR-32 (Fig S7E). On the other hand, as WB in primary tumors showed, kalirin-8 expression was associated with MYCN amplification, but not with MYCN expression, indicating its sub-lineage specificity. RT-PCR for 3′-most exons of kalirin-9 and -12 showed positive results in untreated IMR-32 and SK-N-BE2c (Fig 5H), whereas the full-length kalirin-9 and -12 proteins were barely detectable and a kalirin–GEF2–relevant AB (Fig S5D) did not detect kalirin-8, which implied that other mechanisms (e.g., cleavage by calpains [Miller et al, 2017]) were involved in generating kalirin-8 in NB. Consistently, KALRN gene profiles from primary NB did not support kalirin-8 3′-UTR, which, along with the absence of 5′-most exons of A_23_P307563–detectable Duet and negative RT-PCR results for Duet, indicated the prevalence of kalirin-12 and kalirin-7 in ADRN NB (Fig 5H and I). Given the repertoire of kalirin isoforms in sympathetic neurons (May et al, 2002), we checked KALRN expression in a retinoic acid (RA)-induced neuronal differentiation model, SK-N-BE(2)c, and observed induction of kalirin-9 and kalirin-12 mRNAs and proteins as well as suppression of kalirin-8 after RA treatment. We also noticed TRIO transcripts and proteins' strong induction, whereas TIAM1 mRNA levels remained unchanged (Fig S7F, left and top right). In IMR-32 cells, which are not amenable to differentiation, we observed neither kalirin-8 down-regulation nor induction of kalirin-9, -12, and TRIO (Fig S7F, bottom right).

Taken together, these results suggest that kalirin is involved in RAC1 activation in ADRN-type NB, particularly in its most aggressive, MNA sub-type.

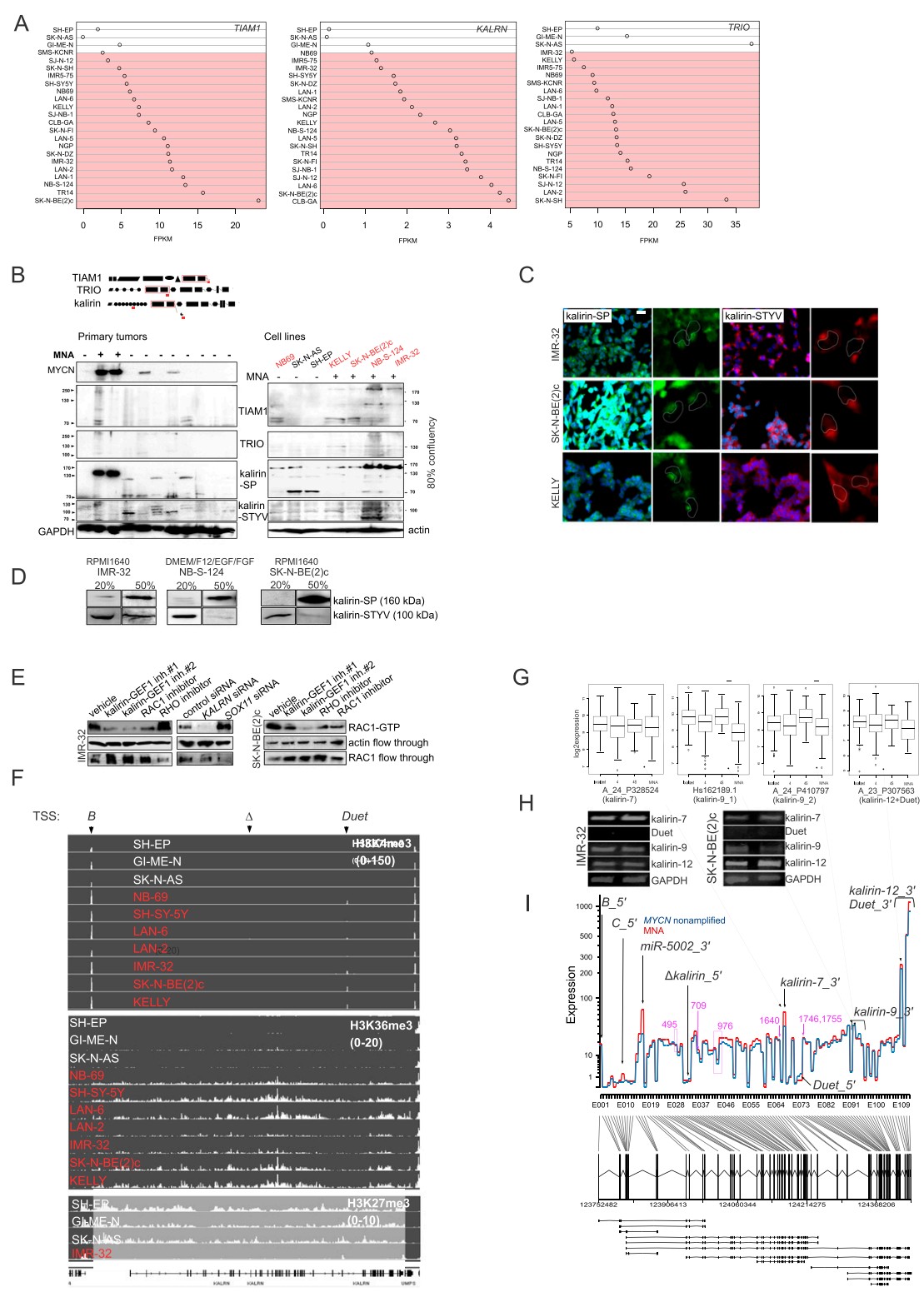

**Figure 5. Kalirin is a NSC23766-sensitive RAC1-GEF in ADRN-type cells.**
**(A)** *TIAM1*, *KALRN*, and *TRIO* mRNA expression in MES and ADRN (marked in red color) cell lines. **(B)** WB analysis of TIAM1, TRIO and kalirin in a panel of primary NBs, NB cell lines. Location of antigens and RAC1-GEF domains in TIAM1, TRIO, and kalirin proteins (top) is marked by red lines and boxes, respectively. MNA status in primary tumors and cell lines is indicated. The names of ADRN cell lines are marked in red color. **(C)** IMR-32, SK-N-BE(2)c, and KELLY were stained with anti-STYV and anti–kalirin–SP antibodies and visualized with Cy3- or Alexa 488–conjugated secondary antibodies. Nuclei are indicated by the dashed lines. **(D)** WB analysis of kalirin–SP and kalirin–STYV in MNA cell lines harvested at 20% and 50% confluency. **(E)** RAC1 activity in IMR-32 and SK-N-BE(2)c cell treated with kalirin–GEF1 inhibitor#1 (10 μM),

## Kalirin inhibition hinders NUC migration, evoking transcriptional signatures associated with low-risk characteristics in primary NB

Kalirin has not been identified as a regulator of NUC or MT function, as its functions are mainly attributed to the regulation of exocytosis and actin cytoskeleton in the post-migratory neurons. Moreover, kalirin is an extremely multifaceted molecule; therefore, we sought to carefully assess cell phenotype after kalirin suppression. Kalirin suppression did not affect cell viability (Fig S7G) and cell cycle distribution in NB cell lines (data not shown), which was in agreement with the pro-migratory role of RAC1. 2D exclusion assay revealed a decrease in migration after kalirin suppression in MNA ADRN cells and, to a lesser extent, in *MYCN*-nonamplified NB69, which was confirmed by cell tracking in IMR-32_NM and NB-S-124 (Figs 6A and B and S8A and B). As cell kymographs and |NC-CC| plotting demonstrated, kalirin-GEF1-suppressed IMR-32_NM displayed cell polarisation defects and NUC defects (Fig 6C and D and Video 7). Immunostaining showed that kalirin was not a centrosomal protein (Fig S6B), but rather colocalised with MTs and Golgi complex, which supported the idea of kalirin involvement in the regulation of the NUC function of MTs (Fig S8C). We noticed that, similar to *DCX*- or *SOX11*-KD, the proportion of the nuclei with distally located centrosomes as well as a variability in the N-C distance were lower in the cells after the suppression of kalirin or RAC1 inhibition (Figs 6E and S8D), indicating that the cells failed to translocate centrosomes, which was in line with the findings in neurons (Tanaka et al, 2004; Yang et al, 2012). Also, inhibition of either kalirin or RAC1 altered γ-tubulin distribution, suppressing extra-centrosomal γ-tubulin signals (Fig S8E). The heterogeneity of nuclear shapes, observed in fixed IMR-32 cells, was decreased in the cells treated with the RAC1 inhibitor or after kalirin suppression, which was an additional indication of a reduction in the cell fraction primed for NUC (Fig 6F and G). We found less distal βIII-tubulin staining in the cells after kalirin suppression (Fig 6H). These alterations closely resembled the defects of nuclear elongation and LP formation, observed after the block of MT polymerization in neurons (Nishimura et al, 2017). Thereafter, we stained IMR-32 for F-actin by fluorescently labelled phalloidin, which revealed a reduction in perinuclear staining in kalirin-GEF1 inhibitor#2-treated cells; therefore, actin regulation by kalirin could not be completely ruled out (Fig S8F). As a further test for kalirin involvement in NUC, we checked whether kalirin–GEF1 inhibition affected the expression of *DCX*, *LIS1* and *SOX11* and found down-regulation of *DCX*, but not *SOX11*, in the kalirin–GEF1–inhibited IMR-32 cells. These results suggested that kalirin suppression could reinforce migration inhibition via *DCX* downmodulation (Fig S8G). We checked whether DCX re-introduction (Tanaka et al, 2004) rescued migration in *KALRN*-KD cells. The substantial compensation of migration was observed in *KALRN*- and *DCX-KD*, but not in *SOX11*-KD IMR32_NM

cells (Fig S8H). The most plausible explanation for this situation is the reprogramming taking place in *SOX11*-KD cells, which makes DCX irrelevant to cell migration. *Kalirin-7* forced expression did not significantly affect motility nor induced differentiation in IMR-32, rescuing migration after application of 5 µM kalirin–GEF1 inhibitor#2 (Fig S9A; data not shown). We resolved migration defects by inspecting NNC/NCC angle distribution in IMR-32_NM after the suppression of RAC1 or kalirin (Fig 6I, left). Less NUC was observed in the RAC1- and kalirin–GEF1–inhibited cells, but NUC impact onto cell velocity was retained after RAC1- or kalirin-GEF1 inhibition (Fig 6I, right and Fig 6J). This, along with the results of |NC-CC| plotting, implied that NUC suppression by RAC1 or kalirin–GEF1 inhibitors did not lead to migration mode reprogramming, which was consistent with the results of RNA-seq in RAC1- and kalirin–GEF1–inhibited IMR-32 (Table S3). Next, we resolved RAC1- and kalirin–GEF1–inhibited IMR-32 cells in the pseudo-3D assay and observed suppressed single cell motility after treatment with either of the inhibitors (Fig 6K). As NC-CC mapping showed (Fig 6I), RAC1- and kalirin–GEF1–suppressed cells retained the impact of negative NC-CC within 0–40° block, which was an indicator of cell contractions. This prompted us to test combination treatments with the ROCK inhibitor and RAC1 or kalirin–GEF1 inhibitors. The treatment with either of kalirin–GEF1 inhibitors or RAC1 inhibitor aided with ROCK inhibitor reduced cell viability in SK-N-BE(2)c (Fig S9B). This effect was not present in other cell lines. In the pseudo 3-D assay, we noticed spheroids' reduced dissociation after the addition of either of the kalirin–GEF1 inhibitors or RAC1 inhibitor to the ROCK-suppressed cells (Fig S9C) or addition of the ROCK inhibitor aided with either the RAC1 inhibitor or kalirin–GEF1 inhibitors (Fig S9D). Our data indicate that double treatments with ROCK- and kalirin–GEF1 inhibitors could reinforce inhibition of migration. The data provided mechanistic evidence for the NUC function of kalirin in ADRN-type NB cells.

Analysis of RNA-seq–resolved profiles of kalirin-GEF1–inhibited IMR-32 showed that the up-regulated genes had an affinity with transcriptomes of stage 4S, *MYCN*-nonamplified tumors and stages 1|2 (Figs 7A and S10A). There was a significant transcriptomic overlap between kalirin–GEF1 inhibition and *DCX* RNAi or RAC1 inhibition in IMR-32, compared with the *SOX11* RNAi (Fig S10B and C; data not shown). Similar to *DCX* RNAi, gene signatures RAC1 neighbourhood, mRNA transport and NMD, MYC(N) and TP53 targets, gene sets pertaining to mitochondrial function and cortical cytoarchitecture were depleted in kalirin–GEF1–inhibited cells (Fig 7B and Table S3). The depletion of TP53 and MYC(N) targets was a trait present under several NUC inhibitory conditions, including *DCX*-KD, *LIS1*-KD, *SOX11*-KD, and kalirin–GEF1–inhibited IMR-32 (Fig S10D); and *TP53* RNAi induced migration defects in IMR-32_NM (Fig S10E), although the *TP53*-KD cells remained adherent (Video 8). The up-regulation of pseudogenes and TCF21-dependent genes was

---

kalirin–GEF1 inhibitor#2 (5 µM), RHOA inhibitor (3 µM), RAC1 inhibitor (10 µM), or after *SOX11* and *KALRN* RNAi. The cell lysates were incubated with RBD–Rhotekin or PAK-PBD beads and the bound activated RHOA and RAC1 was analysed by Western blotting. Flow through fraction was analysed for RHOA and RAC1 expression as loading control. **(F)** H3K4me3, H3K36me3, and H3K27me3 ChIP-seq showing binding events at *KALRN* promoter and gene body in NB cell lines. ADRN cell lines are marked in red color. Arrows indicate transcription start sites. **(G)** Expression of kalirin isoforms in primary NBs. **(H)** RT-PCR for 3′-most exons of kalirin-7, kalirin-12, Duet and kalirin-9 in IMR-32 and SK-N-BE(2)c. **(I)** *KALRN* gene profile obtained from RNA-seq data analysis of 27 primary NBs. 5′-most exons, 3′-most exons of the isoforms are indicated, exons encoding known and high score calpain cleavage sites (predicted for the kalirin-12 [uniprot: O60229-1]) are marked in purple color.

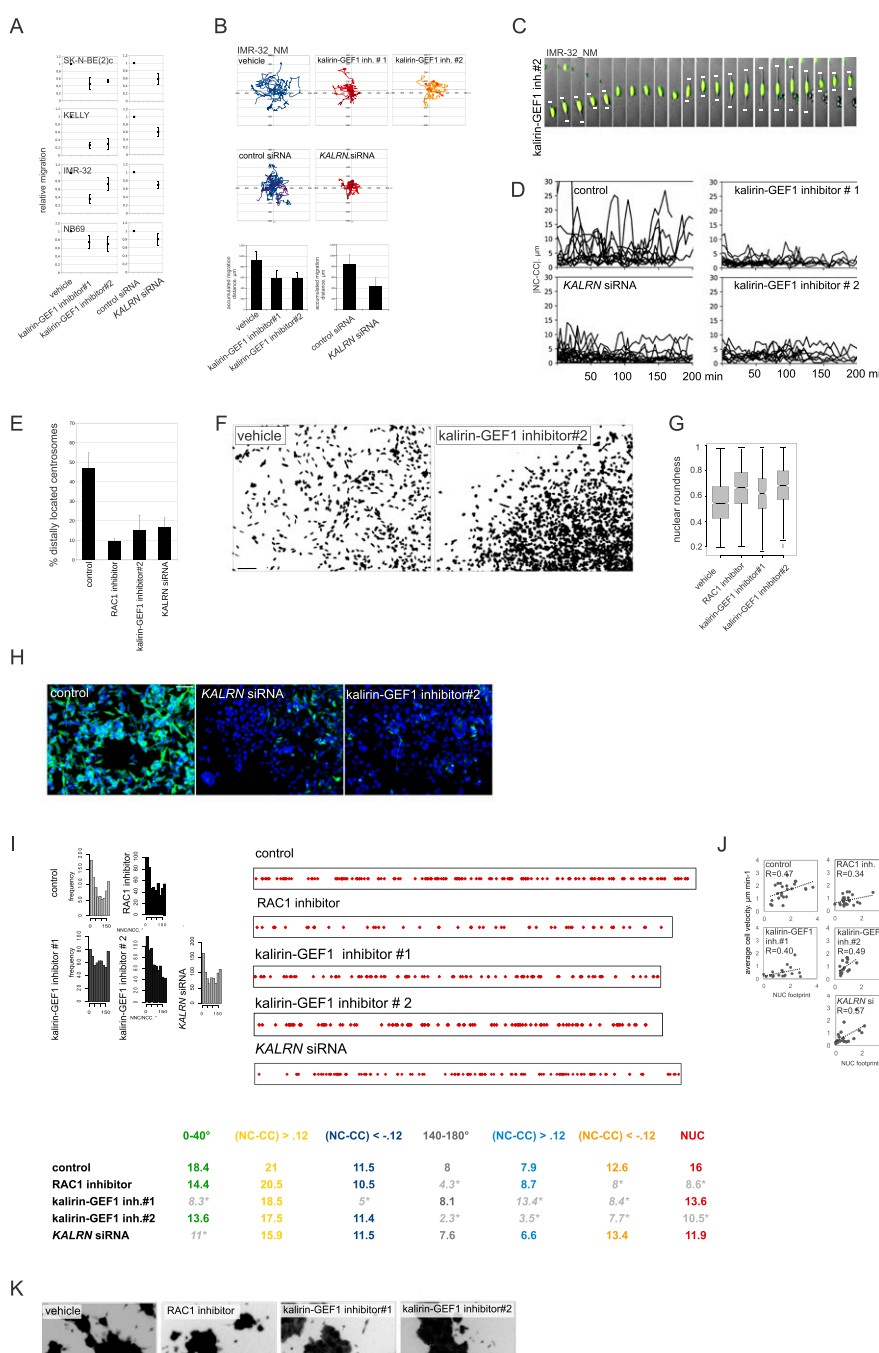

**Figure 6. Kalirin inhibition hinders migration in MNA cells and perturbs cell polarisation and MT structure.**

**(A)** Relative migration in 2D exclusion assay after treatment with vehicle, kalirin–GEF1 inhibitor#1 (10 µM) and kalirin–GEF1 inhibitor#2 (5 µM) or after *KALRN* RNAi. Relative cell migration is quantified via normalization of cell density to vehicle- or siRNA-treated control. Graphs represent mean relative difference in migration + SD. **(B)** Random walk plots and accumulated migration distances in control IMR-32_NM cell after treatment with kalirin–GEF1 inhibitor#1 or kalirin–GEF1 inhibitor#2, and after 48 h of *KALRN* RNAi (13 and 10 h, 15-min intervals). Mean values + SD are presented. **(C)** Time-lapse images of IMR-32_NM after kalirin–GEF1 inhibitor#2 treatment. Nuclei and leading processes are indicated. Scale bar 20 µm. **(D)** |NC-CC| plots in control IMR-32_NM and after treatment with kalirin–GEF1 inhibitor#2, kalirin–GEF1 inhibitor#1 or *KALRN* siRNA. **(E)** The percentage of centrosomes located distally in control, kalirin, or RAC1-suppressed IMR-32 cells. Mean values + SD are presented. **(F)** DAPI staining showing changes in the nucleus shape in IMR-32 after kalirin–GEF1 inhibition. **(G)** Box plots demonstrating nuclear roundness in IMR-32 cells treated with RAC1- or kalirin–GEF1 inhibitor. Data represent three independent experiments (819, 1,008, 375, and 772 cells). **(H)** βIII-tubulin staining in control IMR-32 cells and after treatment with kalirin–GEF1 inhibitor#2 or *KALRN* RNAi. The representative fields were photographed. Scale bar 100 µm. **(I)** NNC/NCC angle frequency distribution (left) and NUC and noise-corrected NC-CC distances in 0–40° and 140–180° signatures in concatenated tracks from control (25 cells and 866 cells), RAC1-inhibited (30 cells, 563 timepoints), kalirin–GEF1–inhibited (25 cells and 606 timepoints) and *KALRN* siRNA treated (30 cells and 860 timepoints) IMR-32_NM (right). **(J)** Correlation plots between cell velocity and NUC footprint in control IMR-32_NM and after treatment with RAC1 inhibitor, kalirin–GEF1 inhibitor#1, kalirin–GEF1 inhibitor#2, or *KALRN* siRNA. **(K)** Phase contrast images, random walk plots, and accumulated migration distances of randomly migrating cells treated with vehicle, RAC1 inhibitor (10 µM), kalirin–GEF1 inhibitor#1 (10 µM) or kalirin–GEF1 inhibitor#2 (5 µM) for 48 h in pseudo-3-D (21 h, 90-min intervals). Mean values + SD are presented.

| | 0-40° | (NC-CC) > .12 | (NC-CC) < -.12 | 140-180° | (NC-CC) > .12 | (NC-CC) < -.12 | NUC |
|---|---|---|---|---|---|---|---|
| control | 18.4 | 21 | 11.5 | 8 | 7.9 | 12.6 | 16 |
| RAC1 inhibitor | 14.4 | 20.5 | 10.5 | 4.3* | 8.7 | 8* | 8.6* |
| kalirin-GEF1 inh.#1 | 8.3* | 18.5 | 5* | 8.1 | 13.4* | 8.4* | 13.6 |
| kalirin-GEF1 inh.#2 | 13.6 | 17.5 | 11.4 | 2.3* | 3.5* | 7.7* | 10.5* |
| *KALRN* siRNA | 11* | 15.9 | 11.5 | 7.6 | 6.6 | 13.4 | 11.9 |

also a trait that RAC1-inhibited and kalirin–GEF1–inhibited IMR-32 shared with the *DCX*-KD cells (Figs 7B and S10F, left). *TCF21* was highly expressed in stage 4S (stage 4S versus stage 4, *P*-value < 0.001; stage 4S versus MNA, *P*-value <0.001; one-way ANOVA test; Fig

S10F, right). Yet, we found neither a difference in *TCF21* expression in the profiles of NUC-suppressed cells nor a possible proxy TF downstream of TCF21, which suggested other regulatory mechanisms behind the TCF21 signature in NB. The transcriptomic overlap

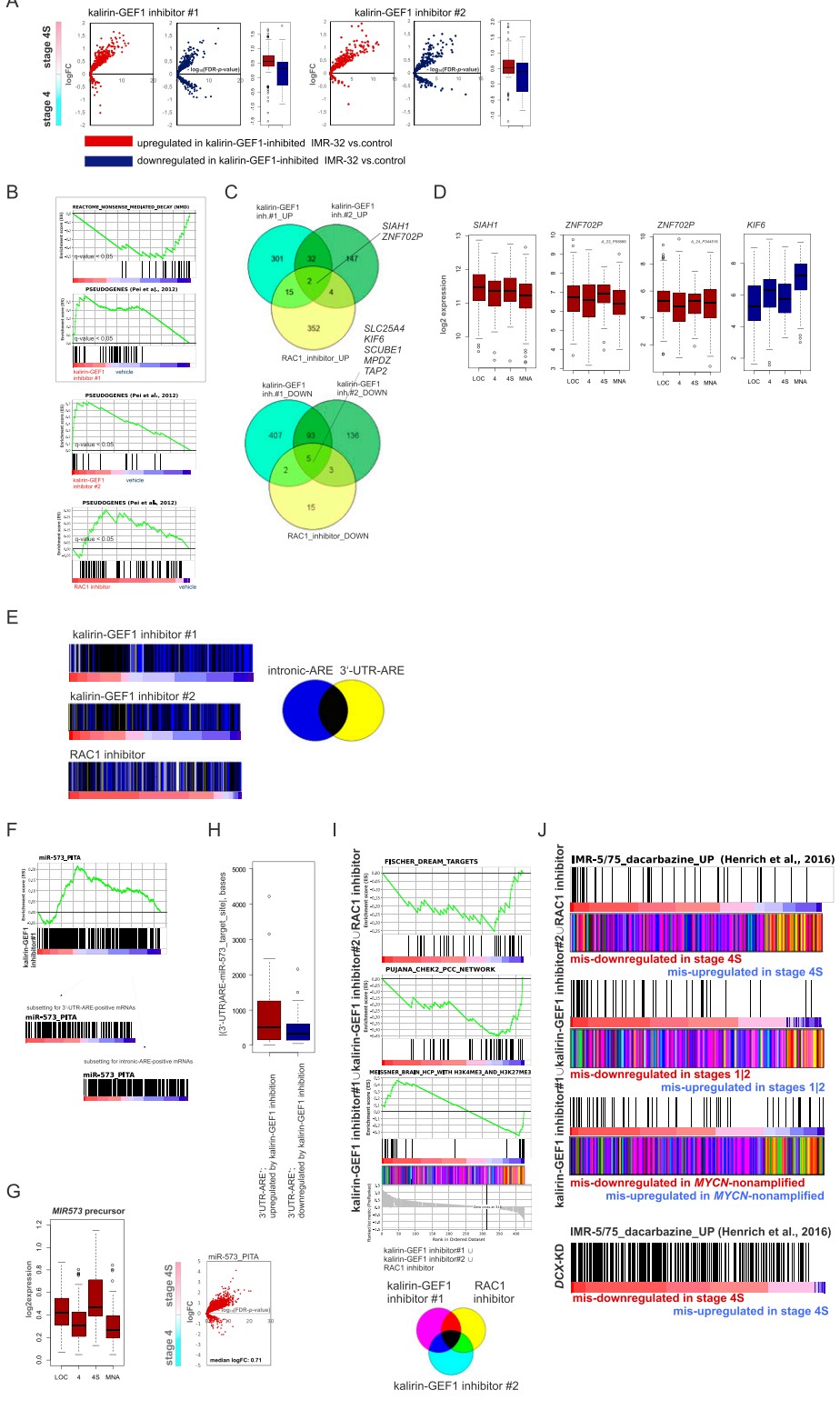

**Figure 7. Inhibition of kalirin-GEF1 engages pathways of post-transcriptional gene regulation.**
**(A)** Volcano plots showing expression of DEGs in kalirin–GEF1–inhibited cells versus control in stage 4S versus 4 tumors (gse49710). **(B)** Gene set enrichment analysis plots showing genes sets with similar pattern of regulation in kalirin–GEF1–inhibited and *DCX*-KD IMR-32. **(C)** Venn diagram showing the number of genes in transcriptomic overlap between DEGs in RAC1 inhibitor-, kalirin–GEF1 inhibitor#1– and kalirin–GEF1 inhibitor#2–treated IMR-32 versus control. **(D)** Box plots demonstrating *ZNF702P*, *SIAH1*, and *KIF6* expression in primary NB tumors. **(E)** Color-coded scheme showing gene overlaps between 3′-UTR- and intronic-AU–rich element (ARE)-containing genes (Bakheet et al, 2018) in the ranked lists of DEGs in RAC1- and kalirin–GEF1–inhibited IMR-32 versus control IMR-32. **(F)** Enrichment plots for miR-573 targets ("miR-573_PITA") based on PITA algorithm (Kertesz et al, 2007) in IMR-32–treated with kalirin–GEF1 inhibitor#1 versus control (top). Diagrams demonstrating miR-573 targets in the ranked gene subsets of 3′-UTR-ARE and intronic ARE containing genes from IMR-32–treated with kalirin–GEF1 inhibitor#1 versus control (bottom). **(G)** Box plot demonstrating *MIR573* expression in primary NB tumors (gse62564) (left) and volcano plot showing expression of predicted miR-573 targets (right) in stage 4S versus 4 tumors. **(H)** Box plot showing distances between AREs and miR-573–binding sites in 3′-UTRs of DEGs in kalirin–GEF1–inhibited cells. *P*-value: 0.005 (two-sample Kolmogorov–Smirnov test). **(I)** Gene set enrichment analysis plots for the indicated gene sets (top) in the combined list of genes mis-regulated (based on their mis-expression in stage 4S versus stage 4 tumors [*P*-values by two-way *t* test ≤0.05; no logFC cutoff]) in RAC1- and kalirin–GEF1–inhibited IMR-32. Color-coded scheme showing gene overlaps (bottom) between mis-expressed genes extracted from profiles of kalirin–GEF1– and RAC1-inhibited cells and ranked according to their expression in stage 4S versus stage 4. **(J)** Diagrams showing positions of the genes up-regulated by dacarbazine in ADRN-type cell lines (Henrich et al, 2016) in the combined lists of genes mis-regulated in RAC1– and kalirin–GEF1–inhibited (top) and *DCX*-KD IMR-32 (bottom).

within the group "RAC1 inhibitor and kalirin-GEF1 inhibitors" (*P*-value cutoff, 0.12; Kruskal–Wallis test) included up-regulation of *ZNF702P*, *ETV1* (lowest *P*-value < 0.02), and *SIAH1* (lowest *P*-value < 0.03), as well as down-regulation of *KIF6*, *SCUBE1*, *EHMT1*, and *MPDZ*

(lowest *P*-value < 0.03) (Fig 7C and Table S3). The expression of *ZNF702P* and *SIAH1* was associated with certain low-risk characteristics in NB (stage 1 versus stage 4, *P*-value < 0.05; *MYCN*-non-amplified versus MNA, *P*-value < 0.05; one-way ANOVA test),

whereas *KIF6* was highly expressed in stage 4 and MNA NB (stage 4 versus LOC, *P*-value < 0.001; MNA versus *MYCN*-nonamplified tumors, *P*-value < 0.001, one-way ANOVA test, Fig 7D). The genes correlated with *ZNF702P*, *SIAH1*, and *KIF6* extracted from the gse49710 dataset showed overlaps with the profiles of kalirin–GEF1– and RAC1-inhibited cells (Fig S11A). As *ZNF702P* was down-regulated in the *SOX11*-KD cells (Table S2), we checked the lineage affinity of *ZNF702P* mRNA, which revealed ADRN type–specific *ZNF702P* expression (Fig S11B). Pseudogene function as miRNA decoys was documented before (Poliseno et al, 2010); therefore, we checked available miRNA–pseudogene interaction data (Li et al, 2014), which revealed the binding of several miRNA species to *ZNF702P* mRNA (Table S4).

Given the regulation of RNA-binding protein-encoding mRNAs in kalirin–GEF1–inhibited cells, cross-checking of our datasets for genes containing AU-rich elements (AREs) (Bakheet et al, 2018) was reasonable, which revealed the prevalence of ARE-containing mRNAs in the up-regulated genes of kalirin–GEF1–inhibited cells (Figs 7E and S11C and Table S4). The enrichment involved mRNAs with 3′-UTR- and intronic AREs. Sub-setting for 3′-UTR-ARE- and intronic ARE-containing genes revealed that they were enriched for the targets of miR-181-5p, miR-153-5p, miR-335-3p, miR-493-5p, miR-12136, and miR-548-5p/3609-3p and the motif "AACTTT_UN-KNOWN" (Table S4). Given the criteria for AACTTT selection (8 kb surrounding a transcription start site; Xie et al, 2005), we reasoned that "AACTTT" motif represented an intronic binding site for a miRNA. miRBase inspection showed that nucleotide string "AACUUU" is potentially targeted by miR-148a-5p, miR-548at-5p, miR-561-5p, and the juxtaposed to miR-12136 sequence "326_104" at 1p36.33, cloned previously from an MNA NB tumor (Afanasyeva et al, 2008). The miRNA precursors, *MIR181A2* and *MIR3609*, were down-regulated in kalirin–GEF1–inhibited cells (Table S3), which could explain the up-regulation of miR-181 and miR-3609 targets. These two miRNA genes, together with *MIR573* and *MIR873*, were present in profile overlaps after at least two of the NUC inhibitory treatments used in this study (*DCX* RNAi, *LIS1* RNAi, *SOX11* RNAi, and RAC1/kalirin–GEF1 inhibition), suggesting essential functions of these miRNAs in NUC regulation. Yet, ARE-positive genes potentially targeted by miR-573 were up-regulated in kalirin–GEF1–inhibited cells (Figs 7F and S11D). A similar effect was visible in *DCX*-KD, after subsetting the profile for ARE-positive genes (Fig S11D). The expression of *MIR573* precursor was higher in stage 4S and LOC tumors (Fig 7G, left; *P*-values < 0.001; one-way ANOVA test). Also, the expression of predicted miR-573 targets showed affinity to stage 4S and stages 1|2. (Fig 7G, right, data not shown). Genes correlated positively with *MIR573* expression in primary NB were enriched for miR-573 targets (Fig S11E). miR-873 targets showed a similar pattern of expression in kalirin–GEF1–inhibitor#1–treated IMR-32, but not in the *DCX*- and *LIS1*-KD cells (Fig S11F). Potential miR-573 binding sites and confirmed experimentally miR-873 binding sites (Li et al, 2014) were localised further from 3′-UTR AREs in the ARE-containing up-regulated genes compared with the ARE-containing down-regulated genes in kalirin–GEF1–inhibited cells (Figs 7H and S11G). Therefore, the up-regulation might require a function in cis of ARE-binding complexes and miRNA-induced silencing complexes (miRISCs). The enrichment for ARE-positive miRNA targets of miR-181-5p, miR-153-5p, miR-335-3p, miR-493-5p, miR-12136, and miR-

548-5p/3609-3p was also apparent in the transcriptomes of stage 4S and stages 1|2, compared with stage 4 primary tumors (Table S4). Similar to *DCX*-KD, kalirin–GEF1 inhibitor[UP] and NB_DTC[UP] gene sets formed an overlap (Fig S11H, left), which was an indication that kalirin–GEF1 inhibition took place in NB in vivo. Also, affinity of miR-573 targets, 3′-UTR- and intronic-ARE–positive mRNAs with transcriptomes of DTCs was observed (Fig S11H, right).

Next, we checked whether and RAC1- and kalirin–GEF1 inhibition affects cell death induced by doxorubicin, paclitaxel, vincristine, and exherin in IMR-32 and SK-N-BE(2)c and observed only minor interaction with the MT drugs, vincristine and paclitaxel (Fig S11I, data not shown). Our data indicated that mis-expression mainly affected down-regulation in RAC1- and kalirin–GEF1–inhibited cells; therefore, possible vulnerabilities were to be found in the mechanisms of down-regulation. We combined mis-expressed genes extracted from the transcriptomic profiles of RAC1- and kalirin–GEF1–inhibited cells and subjected derived lists to GSEA, which revealed the presence of CHEK2 neighbourhood and DREAM targets in a small subset of mis–up-regulated genes (Fig 7I and Table S4). The subset of mis–down-regulated genes was enriched for bivalently marked genes (Fig 7I). This suggested that the activity of the polycomb repressive complex was responsible for the gene down-regulation in the NUC-suppressed cells. Several prospective epigenetic modifiers, including EZH2 inhibitors, were already tested in NB (Henrich et al, 2016; Chen et al, 2018). Yet, the genes up-regulated after treatment of ADRN-type cell line, IMR-5/75, with an EZH2 inhibitor (Henrich et al, 2016) showed little overlap with the profiles of kalirin–GEF1– and RAC1-inhibited IMR-32 (data not shown). On the other hand, the genes up-regulated after the treatment with the DNA-methyltransferase inhibitor (DNMT) dacarbazine showed an overlap with mis–down-regulated genes in kalirin–GEF1– and RAC1-inhibited IMR-32 as well as *DCX*-KD cells (Fig 7J), implying that dynamic DNA methylation might take place in the NUC-suppressed cells. Taken together, these data show that pharmacological inhibition of kalirin–GEF1 evokes transcriptomic traits of low-risk NB.

Our data provide evidence that gene up-regulation driven by kalirin–GEF1 inhibition operates via miRNA-dependent posttranscriptional mechanisms. Most importantly, our analyses suggest that suppressors of epigenetically driven gene down-regulation might interact with kalirin–GEF1 inhibitors synergistically, which should be checked in future experiments.

## Discussion

Intact NUC is necessary for neuronal positioning, but its relevance to cancer, particularly to cancer cell migration, has not been elucidated. In this study, we present evidence that NUC is active in ADRN-type NB. Previous studies revealed the RHOA-ROCK pathway's functionality in NB migration (Matas-Rico et al, 2016). Our study shows that intact migration in ADRN-type NB requires ROCK and RAC1. The interference with the function of RAC1 and the RAC1-GEF, kalirin, leads to severe defects in NUC migration, evoking several transcriptional features of low-risk NB.

Migration in NB is thought to recapitulate certain traits of neural crest-derived cells of origin (Ratner et al, 2016; Delloye-Bourgeois & Castellani, 2018). However, it is unlikely that NUC is active during the

development of sympathetic ganglia. We could not find any information about the involvement of postganglionic sympathetic structures in *Lis1* KO and *Dcx* KO mice. Also, only a few cases have been reported when individuals with lissencephaly had heterotopias and/or hypoganglionosis in the PNS (Mittal et al, 2014). There is one indication that *Lis1* is involved in the tangential migration of certain sympathetic preganglionic neurons (Moore et al, 2012). Evolutionary younger than other neural crest derivatives post-ganglionic sympathetic elements emerge in gnathostoma through phox2, ascl1 and hand coalescence into an expression module (Häming et al, 2011). Therefore, the correspondence between NUC migration in NB and CNS might manifest during the development of the neuronal subtypes expressing NB-like TF code, that is, hindbrain (nor)adrenergic neuronal formations (Zeisel et al, 2018). Particularly, facial branchiomotor neurons demonstrate N-C inversions when migrating tangentially (Distel et al, 2010). Also, motor neurons migrate tangentially in a reelin-negative region (Rossel et al, 2005), which is an interesting notion when put in the context of *RELN* negativeness of advanced NB (Becker et al, 2012). Probably, migrating ADRN NB resembles motor neurons, but "gets stuck" at the stage of tangential migration-like centrosomal repolarisation. Remarkably, murine orthologue of up-regulated in kalirin–GEF1–inhibited cells *ETV1* is expressed in the nuclei of mouse cranial nerves (Zeisel et al, 2018). More precisely, Etv1 expression appears in facial motor neurons during the final post-tangential stage of their migration, the sub-nuclear segregation, and is indispensable for finalising neuronal differentiation (Zhu & Guthrie, 2013; Tenney et al, 2019). Kalirin–GEF1 inhibition up-regulates low-risk specific transcriptomic traits in ADRN NB, which can be interpreted as a sign of a differentiation-like process. Yet, it is the kalirin paralog, Trio, that is involved in facial motor sub-nuclear segregation in mice (Backer et al, 2007). On the other hand, in xenopus, *kalrn* expression is present in the cranial nerves, whereas *trio* is expressed in migrating neural crest cells (Kratzer et al, 2019). In fact, kalirin and trio originated through the duplication of a proto-trio/kalirin gene in the ancestral invertebrate, but became first stabilised and functionally diversified in cyclostomata (Kratzer et al, 2019). Probably, kalirin-dependent NUC migration in ADRN NB recapitulates the relict hindbrain migration. It would thus be interesting to assess neuronal migration in the hindbrain of *kalrn* morphants.

Given the parallelism between neuronal and NB migration, it is reasonable to check if there have been any studies reporting the co-occurrence of NB and neuronal overmigration in the CNS. Indeed, a possible manifestation of neuronal overmigration, cortical polymicrogyria (Squier & Jansen, 2014), is found in Weaver syndrome, which is caused by mutations in the *EZH2* gene (Tatton-Brown et al, 2013), and in a neuroactive drug embryopathy, fetal hydantoin syndrome (Al-Shammri et al, 1992). Importantly, these disorders rarely involve only the forebrain, but also manifest in hindbrain malformations. Both of these conditions are associated with a higher risk of NB. Interestingly, two NB cases reported in children with Weaver syndrome were stage 4S tumors. Also, some NB tumors that were observed in children with fetal hydantoin syndrome were diagnosed perinatally (Al-Shammri et al, 1992), which may be indicative of low-risk stage tumors. Remarkably, polymicrogyria loci include 1p36 (Dobyns et al, 2008) which is frequently deleted in NB tumors (White et al, 2001). A small percentage of children with constitutive 1p36 monosomy developed NB tumors which regressed spontaneously (Biegel et al, 1993; Isidor et al, 2008). It is tempting to speculate that the stage 4S NB also stems from being ectopically activated and supported by fetally expressed chemokines NUC migration rather than neural crest-like forms of migration. Furthermore, NUC deactivation via cell-non-autonomous events (e.g., via fetal to neonatal transcriptome switches that affects NB homing) may trigger programmed cell death in stage 4S tumors. Therefore, the functional assessment of polymicrogyria candidate genes, for example, 1p36 candidates, α-enolase encoding *ENO1*, and arginine–glutamic acid dipeptide repeats protein-encoding *RERE* (Jordan et al, 2015; El Waly et al, 2020), in the context of NB can be very informative.

Similar to postganglionic sympathetic structures, the hindbrain is rarely involved in classic forms of lissencephaly (Jissendi-Tchofo et al, 2009), which is, in part, caused by the functional redundancy of NUC regulators. In line with that, double *Dcx*, *Dclk1* KO mice have severe cerebellar and brainstem defects and resemble *Cdk5* KO (Ohshima et al, 1996; Deuel et al, 2006). Because of the function of NUC genes in differentiation and migration, migration in ADRN NB cells likely relies on a minimal set of NUC genes. Particularly, LIS1 that normally controls spindle assembly in NE is not involved in proliferation maintenance in ADRN NB, which supports the idea of NUC gene module. NUC activity in G1, migration inhibition in *TP53*-KD cells and down-regulation of TP53 targets in NUC inhibited cells suggest that NUC migration is intertwined with the control of cell cycle transition in NB. TP53 regulates genes encoding NUC-controlling cyclin inhibitors (Kawauchi & Nabeshima, 2019) such as CDKN1C (p57kip), and genes involved in synapse maintenance (Merlo et al, 2014). Given the low frequency of *TP53* mutations in NB, it is plausible that a part of p53 activity is rerouted to the NUC program. On the other hand, *TP53*mut ADRN NB cell lines are also affected by NUC. How *TP53*mut cells bypass TP53 control of NUC also remains to be answered. In addition, DCX and LIS1 functions are not confined to NUC because of their involvement in MT transport and regeneration in PNS (Nawabi et al, 2015; Hines et al, 2018). The defects in LP and lack of significant alterations in N-C distance, which we observed in *DCX*-KD cells, conform with this idea. Also, ADRN NB terminals are reminiscent of pheochromocytoma' *varicones* (Mingorance-LeMeur & O'Connor, 2009), which are akin to exocytotically active postsynapses. So far, nothing is known about their propensity to transduce pro-survival and cell death signals to the nucleus. We can speculate that MT dysfunction in the terminals of NUC-inhibited cells leads to the collapse of pro-survival retrograde signals, which manifests in the down-regulation of metabolic signatures. In line with that, the shut-down of mitochondrial function has been noticed previously upon KD of *Dcx* paralog, *Dclk1*, in mouse NB cells (Verissimo et al, 2010). The N-C inversion mode of NUC observed in NB cannot be explained by dynein-dependent forces (Sakakibara et al, 2013), implying the involvement of actin-based forces. Nevertheless, in ADRN NB, MT function lies upstream of actomyosin forces. MT-binding proteins such as DCX can serve as bridges between MTs and actin in neurons (Nawabi et al, 2015); MT destabilisation in ADRN NB cells might disrupt these links, crushing the entire migration machinery. The N-C-inversion mode requires the concerted action of process maturation and dynamic N-C

attachment. Most likely, kalirin–GEF1 inhibition locks centrosomes in an attached state.

Whereas neurons are not able to activate safeguard migration modes, cancer cells frequently undergo transitions from one migration type to another. The RNAi of a regulator of NUC in the CNS, *SOX11*, triggers NUC-to-MES transition in ADRN-type cells. Yet, the reprogramming is not limited to the migration mode but also affects cell identity, manifesting in ADRN-to-MES transition. The lack of direct binding of SOX11 to *DCX* promoter makes reasonable a search for neuron-specific, SOX11-responsive TFs that regulate the *DCX* promoter (Piens et al, 2010). We cannot conclude whether the regulation of DCX "by proxy" is inherited from a neural crest precursor or appear during ADRN NB oncogenesis. Particularly, in ADRN NB, apart from *DCX*, SOX11 might be ousted from other "legal" locations by MYCN (Zeid et al, 2018). This may explain why previously identified SOX11 targets do not show statistically significant depletion in *SOX11*-KD cells. The effects of *SOX11*-KD can be partially explained by the down-regulation of SOX11-dependent *ELAVL2*, which positively regulates the stability of mRNAs in neurons (Scheckel et al, 2016). Also, ADRN-to-MES transition after *SOX11* RNAi can be caused by the asymmetric distribution of mRNAs encoding ADRN determinants, rather than by withheld transcriptional control of these determinants. Recently, the identity-affecting asymmetric distribution of lysosomes and nuclear promyelocytic leukemia bodies triggered by nuclear migrations was observed in keratinocytes (Lång et al, 2018). A similar process might take place in *SOX11*-KD cells, as nuclei show a tendency towards apical localisation in *SOX11*-KD cells. *SOX11* RNAi may thus eliminate "carriers" of ADRN determinants, whereas not affecting the mesenchymalised counterparts, which would explain the compensated shedding of pyknotic cells that was visible in *SOX11*-KD IMR-32. The best way to test this idea is through live imaging with to-be-designed ADRN and MES sensors. Also, if this idea is true, SOX11-suppressed cells should be an amalgam of transitory states; it is worth it, therefore, to resolve the *SOX11* RNAi population using single-cell sequencing. Importantly, SOX11 down-modulation and EMT induction do take place after chemotherapy in NB. Hence, asymmetric divisions, if their association with the generation of MES cells is proven in vivo, may be an unwanted consequence of certain chemotherapy strategies in NB.

Remarkably, none of MES "heavy artillery" SE-associated TFs, such as *PRRX1* (van Groningen et al, 2017), are induced after *SOX11* RNAi. *Prrx1*-expressing mouse mesenchyme separates from a bipotent autonomic-mesenchymal precursor (Soldatov et al, 2019). None of ADRN-to-MES transitions identified so far, except for *PRRX1* enforced expression, evoked *PRRX1* expression, and events like *SOX11* KD may generate an intermediate cellular compartment that can "descend" to a PRRX1⁺ MES precursor. In mice, Sox11 mRNA is present in the bridge cell population that connects Schwann cell precursors with chromaffin cells (Furlan et al, 2017) and is indispensable for the proliferation of tyrosine hydroxylase-expressing precursors in developing sympathetic ganglia (Potzner et al, 2010). It would be interesting to test whether ectopically asymmetric cell divisions are related to Coffin-Siris syndrome's association with schwannomatosis (Schrier et al, 2012). Following this idea, an efficient NUC inhibitor should suppress migration without hampering the symmetricity of divisions, which adds complexity to the concept of separate targeting of proliferation and migration (Brabletz, 2012). Pending future mouse experiments, it is important to mention that NB DTC (Rifatbegovic et

al, 2018) transcriptomes are recapitulated by *DCX* RNAi and kalirin-GEF1 inhibition. It is tempting to speculate that the fingerprints in DTC profiles formed by *DCX*-KD and kalirin-GEF1 inhibitors reflect DTC heterogeneity and originate from regression events.

Importantly, ADRN NB cells treated with different kalirin–GEF1 inhibitors develop several overlapping gene expression patterns, e.g., down-regulation of NMD gene signature. NB is renowned for its low mutational load (Pugh et al, 2013); therefore, the reason for NMD maintenance in NB should lie outside the concept of NMD-driven degradation of mutated tumor suppressor-encoding RNAs (Popp & Maquat, 2018). We believe NMD inhibition is linked to pseudogene RNA boosting in kalirin-GEF1-inhibited NB cells. It is plausible that transcribed pseudogenes function as miRNA decoys in NB cells. Yet, the expression of matching protein-coding counterparts is not affected by kalirin–GEF1 inhibition, suggesting the involvement of secondary miRNA targets. Particularly, this pertains to the top candidate, human-specific *ZNF702P* at 19q13.4. As ENCORI database mining reveals, *ZNF702P* RNA binds NB metastasis-associated miR-181a-5p and miR-23a-3p (Cheng et al, 2014; Gibert et al, 2014; Liu et al, 2018). Thus, *ZNF702P* can be responsible for the up-regulation of miR-181a targets in kalirin-GEF1-inhibited cells. A miR-181a precursor, *MIR181A2*, was down-regulated in kalirin–GEF1–inhibited cells, which cannot be explained by a direct target-induced miRNA degradation (Haas et al, 2016). Also, *MIR181* pops-up in several transcriptomic profiles in this study, suggesting an essential role for miR-181a in ADRN NB migration. Importantly, these observations have parallels in neurons. NMD is an important mechanism of axon pathfinding and synapse maintenance (Long et al, 2010; Colak et al, 2013), and its defects have resulted in several neurodevelopmental illnesses (Jaffrey & Wilkinson, 2018). Also, a recent study showed that pseudogenes' competition with coding genes for miRNAs has functional relevance in the CNS (Barbash et al, 2017). Yet, pseudogenes are mostly non-conserved, which also pertains to *ZNF702P* and other pseudogenes that are regulated by kalirin-GEF1 inhibition. *ZNF702P* possibly adopts a decoy role from a conserved counterpart. Also, *ZNF702P* is expressed not only in the hindbrain, but also in macrophages and lymphocytes, suggesting a broad range of its functions. Furthermore, the expression of the kalirin-9 isoform with a stop-codon carrying cryptic exon can be explained by the concerted action of two types of machinery: NMD inhibition and alternative splicing. The latter can, in turn, explain high kalirin-9 expression in stage 4S tumors that may have consequences in overall RAC1 activity in NB in vivo (Deo et al, 2012).

Several genes that are down-regulated by kalirin-GEF1/RAC1 inhibition cannot be assigned to a particular gene signature. One such "orphan" gene, kinesin family member *KIF6* at 6p21.2, is thought to influence cilia function (Konjikusic et al, 2018). Cilia support neuronal migration (Higginbotham et al, 2012) in the CNS via conducting SHH signals (Baudoin et al, 2012). In primary NB, *KIF6* mRNA correlated genes contains few MT signatures, being enriched for the genes pertaining to ER-mitochondria-peroxisome nexus. Down-regulation of mitochondrial signatures repeatedly appears in NUC-inhibited cells. *KIF6* down-regulation might eliminate a metabolic add-on that is indispensable for the migration. Another promising candidate, Kleefstra syndrome–associated euchromatic histone lysine methyltransferase 1 *EHMT1* at 9q34.3, encoding a SOX11-interacting methyltransferase (Heim, 2014). Kleefstra syndrome, a neurodevelopmental disorder characterised by impaired

memory, autistic features and intellectual disability (Benevento et al, 2016), shares several clinical traits with an SOX11 deficiency disease: Coffin-Siris syndrome. Intriguingly, abrogation of EHMT1 activity in neurons results not only in anticipated gene expression up-regulation but also in H3K9me3-mediated down-regulation of a gene subset, including clustered protocadherins (Iacono et al, 2018). It would be interesting to test whether loss of EHMT1 occupancy triggers an invasion of EZH-driven heterochromatinisation machinery in NB cells.

When compared with the mRNAs down-regulated in kalirin–GEF1–inhibited cells, the up-regulated mRNAs do not appear to be organised into expression signatures, while showing strong expression affinity to NB subsets with low-risk characteristics. We reasoned that posttranscriptional mechanisms might be a clue to the up-regulation mechanism, which led to a rather serendipitous identification of 3'-UTR- and intronic ARE elements bearing genes up-regulation in kalirin–GEF1–inhibited cells. The mechanism behind this enrichment remains to be identified. We believe this up-regulation is afflicted by miRNA-induced induction that takes place when AREs and MREs co-occur (Vasudevan & Steitz, 2007). *MIR573* at 4p15.2 and *MIR873* at 9p21.1 are candidate "MRE-code" miRNAs that may act in an antagonistic way in NB depending on the status of miRISC complex. Importantly, the list of these "switcheroo" miRNAs does not include miR-16 that targets ARE motif directly (Jing et al, 2005). When mapped to the profiles of kalirin–GEF1–inhibited cells, up-regulated "ARE+; MRE-code+" mRNAs form dense "fingerprints" with moderately high median FC values. Similar fingerprints are also identifiable when profiles from kalirin–GEF1–inhibited cells are mapped onto expression profiles from NB DTC as well as stage 4 versus 4S expression dichotomies. These findings provide strong evidence that kalirin–GEF1 inhibition occurs in NB in vivo. It is known that weak repression by miRNAs can nevertheless have a substantial effect on cell phenotype (Flynt & Lai, 2008). Recently, data have been obtained supporting an FC threshold as low as 1.3 for functional miRNA targets (Yoon et al, 2019). This threshold can apply to the miRNA-induced mRNA up-regulation. Nevertheless, we could not find any perturbations in the expression of miRISC-encoding mRNAs in kalirin–GEF1–inhibited cells, which suggests that other mechanisms (i.e., posttranslational modification) are involved in miRISC regulation. Indeed, the function of a component of miRISC, fragile X mental retardation syndrome-related protein 1, FXR1, is influenced by p21-activated kinase, PAK1 (Say et al, 2010), which, in turn, is potentially activable by kalirin–RAC1. These findings are also a plausible explanation as to why kalirin proteins are consistently found in the nuclei of NB cells. Importantly, gene up-regulation and translation activation by miRNAs are normally observed upon G1/G0 growth arrest and associated with quiescence and differentiation (Vasudevan & Steitz, 2007); therefore, the enrichment for ARE-MRE containing mRNAs is likely a remnant of a differentiation program. Whether or not this phenomenon occurs specifically in regressing versus advanced NB tumors should be clarified in further research.

The administration of anti-migration drugs is devised to be continuous; therefore, the requirements for low toxicity of migration blockers are very stringent (Gandalovičová et al, 2017). As our experiments demonstrate, kalirin–GEF1 inhibitors fit this criterion in vitro. Also, studies with KO mice demonstrate that Kalrn is dispensable for CNS and PNS development (Mandela et al, 2012), which provides additional support for kalirin as a suitable target for migration blockers in NB. Indeed, in the CNS, errors in migration resulting in ectopic neurons are not uncommon as demonstrated by mice KO for Bcl-2–associated X protein-encoding, *Bax* (Jung et al, 2008). However, in sympathetic ganglia of *Bax* KO mice, increased neuronal survival in situ rather than gross ectopias are observed (Deckwerth et al, 1996). Kalirin suppression with an antisense RNA does not affect the survival of mature sympathetic neurons (May et al, 2002). Yet, we still cannot exclude the possibility that kalirin–GEF1 inhibition is involved in programmed cell death in stage 4S NB cells, being insufficient to purge those NB cells that harbour secondary alterations (e.g., *MYCN* amplification). This complete uncoupling from proliferation together with the observation that mis-expression induced by kalirin-GEF1 inhibition mainly concerns down-regulated genes allow us to consider prospective synthetically lethal (Nijman, 2011) KALRN interactions. More than 10% of these "mis-expressed" genes were identified previously as up-regulated by DNMT1 inhibition in ADRN NB (Henrich et al, 2016), suggesting that epigenetic drugs (DNMT- and EZH2 inhibitors) warrant further evaluation in the context of kalirin–GEF1–inhibited ADRN-type cells. Whether the combination of kalirin–GEF1 inhibitors with the drugs has a synergistic effect should be explored in a physiologically relevant environment. Because many human-specific transcripts are likely to impact the phenotypes we observed, we prioritise organotypic 3-D cell culture and tumor slice culture (Sivakumar et al, 2019) as well as ex vivo DTC treatment for further drug screening. Finally, we think that NB is not the only cancer type affected by NUC. Small cell lung cancer, pheochromocytomas and medulloblastomas express genes required for NUC migrations and might also spread nucleokinetically.

# Materials and Methods

### Patients

Neuroblastoma tumor samples were collected before any cytoreductive treatment, snap-frozen, and stored at −80°C until RNA or DNA isolation. Written informed consent was obtained from patients' parents for tissue sampling. Genomic *MYCN* status was assessed in the reference laboratories of the German Neuroblastoma trial in Cologne and Heidelberg.

### Antibodies and reagents

Rabbit anti-kalirin-SP antibody (MBS821543), goat-kalirin-STYV antibody (ab52012), and rabbit kalirin–GEF2 antibody (TA590559) were purchased from MyBioSource, Abcam, and Acris. Rabbit anti-TRIO antibody (A304-269A) was from Bethyl Laboratories, rabbit anti-TIAM1 antibody (ST1070) was from Calbiochem, rabbit anti-SOX11 was from Millipore, mouse (GTU88), and rabbit anti-γ-tubulin (DQ-19) antibodies were from Sigma-Aldrich. The rabbit polyclonal antibody, SOX11-PAb, was custom made (Absea Biotechnology) against the immunogenic peptide p-SOX11[C–term] DDDDDDDDDELQLQIKQEPDEEDEEPPHQQLLQPPGQQPSQLLRRYNVAKV-PASPTLSSSAESPEGASLYDEVRAGATSGAGGGSRLYYSFKNITKQHPPPLAQ-PALSPASSRSVSTSSS (Decaesteker et al, 2020 *Preprint*). Mouse

anti-Golgi apparatus (NB37-100) was from Merck Millipore. Rabbit anti-RAC1/2/3 (2465), rabbit anti centrin-2 (2091), rabbit anti-DCX (4604), and rabbit anti-βIII-tubulin (D71G9) antibodies were purchased from Cell Signalling. Mouse anti-RAC1 antibody (#ARC03) was from Cytoskeleton. Protein G-Agarose was purchased from Santa Cruz Biotechnologies (Protein A/G-Agarose; sc-2003). β-actin antibody and sheep anti-goat HRP-conjugated antibody were from Santa Cruz Biotechnologies. HRP-conjugated antibodies against mouse or rabbit and the Cy3-conjugated anti-goat antibody were obtained from Jackson Immunoresearch Laboratories. The Alexa 488–conjugated goat anti-rabbit antibody (ab150077) and Cy5.5-conjugated goat anti-mouse (ab6947) were obtained from Abcam. RA and Calcein AM were purchased from Sigma-Aldrich. RAC1 inhibitor NSC23766 and cytochalasin B were purchased from Sigma-Aldrich. TRIO/kalirin-GEF1 inhibitors, ITX3 (2-[(2,5-Dimethyl-1-phenyl-1H-pyrrol-3-yl)methylene]-thiazolo[3,2-a]benzimidazol-3(2H)-one), and NPPD (1-(3-nitrophenyl)-1H-pyrrole-2,5-dione), were purchased from Sigma-Aldrich and Matrix Scientific, respectively. RHOA inhibitor, rhosin, was from Sigma-Aldrich. All the reagents were first dissolved in the medium before application to the cells. ROCK inhibitor, Y27632, was obtained from StemCells Technologies. Colcemid was from Thermo Fisher Scientific.

## Cell culture and cell viability assay

Neuroblastoma cell lines were maintained in RPMI1640 supplemented with 10% fetal bovine serum. NB-S-124 cell line was established from infiltrated bone marrow aspirate from 1-y-old patient (established by Dr. F Westermann; Lodrini et al, 2013) and was grown in DMEM/F12 (Thermo Fisher Scientific) with 6% NeuroCult SM1 Neuronal Supplement (StemCells Technologies), 20 ng/ml bFGF (Promocell) and 5 ng/ml EGF (Promocell) at 37°C, 5% CO$_2$ using gelatin-coated cellware. Spheroids were generated by coating 96 well flat bottom plates with 1% agarose. 5,000 cells were seeded per each well and allowed to form spheroids for 72 h. For cell viability Alamar Blue (AbD Serotec) was used according manufacturer's instructions. Fluorescence was detected using the FluorStar Optima microplate fluorescence reader (BMG Labtech).

## DNA constructs, siRNAs and siRNA transfection

Human *kalirin-7* (NM_003947) pENTR223.1 entry clone was retrieved from DKFZ clone repository (DKFZ). *Kalirin-7* cDNA was subcloned using Gateway technology (Thermo Fisher Scientific) into the pTREX31 vector (Thermo Fisher Scientific). Stable SOX11 expression was achieved with transducing SOX11 cDNA sequences were into pLenti6.3/TO/V5-Dest (Thermo Fisher Scientific) and pLVX-Tet3G (Clontech). For siRNA transfection, cells were seeded in six-well tissue culture plates 24 h before transfection. 100 nM of siRNA non-targeting control (siRNA NTC; Dharmacon) or siRNA *SOX11* (Dharmacon) were transiently transfected using DharmaFect 2 (Thermo Fisher Scientific) according to the manufacturer's guidelines. SOX11 expression was induced by doxycycline addition (0.1 µg/ml); and induction was assessed by WB for SOX11 (Decaesteker et al, 2020 *Preprint*). mCherry-gamma-tubulin-17, CyPet-RAC1, and DCX-RFP (Tanaka et al, 2004) constructs were from Addgene repository. pFUCCI G1 Orange construct was from MBL. For knockdown experiments with siRNAs (20 µM stock; 10 nM final concentration),

Lipofectamine RNAiMAX reagent (Life Technologies) was used according to manufacturer's instructions. *KALRN* (sc-18592), *LIS1* (sc-35814), *DCX* (sc-35214), *HNRNPK* (sc-38282), and *DISC1* (sc-60539) were purchased from Santa Cruz Biotechnologies. DCX siRNA used in the DCX-RFP experiments was from Thermo Fisher Scientific. *RAC1* siRNAs were prepared by chemical synthesis (Sigma-Aldrich) using sequences provided by Kutys and Yamada (2014). *ROCK1* siRNA (SIHK1980; SIHK1981) and *ROCK2* siRNA (SIHK1983; SIHK1984) were purchased from Sigma-Aldrich, SOX11 siRNA was from Thermo Fisher Scientific. Transfection with a Silencer Select Negative Control siRNA #1 (Thermo Fisher Scientific) was used as controls.

## Real time quantitative reverse transcription PCR (qRT-PCR) and reverse transcription RT-PCR

Real-time qRT-PCR was performed by using the Applied Biosystems 7000 Sequence Detection system (Applied Biosystems). Amplification of cDNA by real-time PCR was quantified using SYBR Green (Thermo Fisher Scientific) as previously described (Afanasyeva et al, 2011) with the following QuantiTect Primer Assay: *LIS1* (QT00013447), *DCX* (QT00008540) and *SOX11* (QT00221466). *ROCK1* primers were retrieved from PrimerDB (Pattyn et al, 2003). *ROCK2* primers were as designed by Li et al (2015). RT-PCR for kalirin-12, kalirin-9, and DUET 3′-UTR was performed as described previously (Mains et al, 2011) using the primers homologous to mouse *Kalrn*.

## cDNA sequencing

PCR fragments were purified using the QIAquick PCR Purification Kit (QIAGEN). Sequencing of RT-PCR products was performed with primers used in the PCR reaction (Mains et al, 2011). FinchTV (Geospiza Inc.) was used to visualize electrophoregrams.

## 2D migration (exclusion) assay

Neuroblastoma cells were seeded into 96-well plates (Platypus) coated with fibronectin/collagenIV and grown to confluence. A single wound was then created in the cell monolayer and migration of the cells from the edge of the wound was analysed. The area between the wound edges was measured using ImageJ software (National Institutes of Health; NIH). The area of the wound in the control cells was set as 100%, and the relative change was calculated as a percentage of the initial area.

## Time-lapse imaging

Time-lapse images of living cells were captured using Ti-e Eclipse (Nikon) equipped with an incubation chamber. Neuroblastoma cell lines were cultured in RPMI-1640 medium (Thermo Fisher Scientific) as described above on plastic-bottomed chamber (Ibidi) pre-coated with Collagen IV/Fibronectin/Laminin (Sigma-Aldrich). Cells were analysed with a 10×, 20× and 40× objective lens. Optimal time-lapse intervals (IMR-32; 5 min; median velocity: 0.91 µm/min; SH-EP; 15 min; median velocity: 0.3 µm/min) were chosen to capture nucleokinetic or mesenchymal migration. Imaging data were stored as files in NIS-Elements software (Nikon). Object segmentation was performed using ilastik (Sommer et al, 2011). The time- intervals

were assembled by ImageJ and analysed using "Manual Tracking Tool," "Nucleus J" and "Mosaic" plugins. For noise correction of NUC events, the median value of 0.12 $\mu$m derived from analysis of nuclei in sessile IMR-32 cells (migration speed < 0.05 $\mu$m/min) was implemented.

## GTPase pulldown

Rho GTPase activation was analysed by a modification of a protocol described previously (Ren et al, 1999). Cells were washed with PBS, placed on ice, and scraped into lysis buffer (25 mM Hepes, pH 7.3, 150 mM NaCl, 100 mM NaCl, 2 mM $MgCl_2$, and 1% Nonidet P-40 supplemented with protease inhibitors cocktail) and then lysed. Insoluble material was removed by centrifugation for 10 min at 9,500$g$. Lysates were incubated with 1 $\mu$g of PAK-PBD or Rhotekin-RBD (Cytoskeleton) for 16 h at 4°C. Beads were washed, and the proteins were eluted with Laemmli sample buffer and analysed by immunoblotting with anti-RAC1 antibody according to the manufacturer's instructions.

## Immunocytochemistry

Cells growing on slides were fixed with 4% paraformaldehyde in PBS or ice-cold methanol for tubulin/Golgi Apparatus-compatible staining, washed with PBS, and permeabilized with 0.1% Triton X-100. After blocking with 5% fetal bovine serum and 1% BSA, the slides were incubated with primary antibodies against kalirin-PH and kalirin-STYV, TRIO, TIAM1, $\beta$III-tubulin, $\gamma$-tubulin, and Golgi apparatus, washed with PBS and incubated with secondary antibodies: Cy3-conjugated donkey anti-goat, Alexa Fluor 488–conjugated sheep anti-rabbit or Cy5.5–conjugated goat anti-mouse. Triple immunostainings with anti–kalirin–STYV antibody were processed with Cy3-conjugated donkey anti-goat antibody first. Alexa Fluor 555–phalloidin staining was performed according to the manufacturer's instructions (Thermo Fisher Scientific). The slides were counterstained with DAPI (1 $\mu$g/ml) for 5 min. Phalloidin-stained cells were photographed within 30 min after the staining procedure.

## Western blotting

Western blotting was performed as described previously (Afanasyeva et al, 2011). Briefly, the whole cells were prepared in a buffer containing 7 M urea, 1% Triton X-100, 100 mM DTT, 20 mM Tris–HCl, pH 8.5. Protein concentrations were determined by Bradford assay (Bio-Rad), and 50 $\mu$g protein lysate were separated per lane on either 7.5% or 12% PAGE gels then transferred to nitrocellulose membranes (Protran). Membranes were incubated with the appropriate antibodies, and the bands were visualized using the ECL system (Pierce). Images were captured with a CCD camera (Vilber Lourmat).

## Chromatin immunoprecipitation DNA-sequencing (ChIP-seq) of histone modifications

Formaldehyde cross-linking of cells, cell lysis, sonication, chromatin immunoprecipitation (IP) procedure, and library preparation were performed as described previously (Blecher-Gonen et al, 2013), starting with ~4 × 10$^6$ cells (1 × 10$^6$ cells per individual IP). Direct cell lysis for each sample was achieved by incubation for 30

min in 950 $\mu$l RIPA I on ice (10 mM Tris–HCl, pH 8.0, 1 mM EDTA, pH 8.0, 140 mM NaCl, 0.2% SDS, and 0.1% DOC). Tissue disruption, formaldehyde fixation, and sonication of tumor material were performed according to a previously published protocol (Dahl & Collas, 2008). Approximately 30 mg of fresh-frozen tumor tissue was used per individual ChIP-seq experiment. All subsequent steps were performed analogous to cell line experiments. The Bioruptor Plus sonication device (Diagenode) was used for high intensity sonication for 30–60 min each with 30 s on and 30 s off intervals. For the IP antibodies for H3K4me3 (#ab8580; Abcam), H3K36me3 (#ab9050; Abcam) and (#39155; Active Motif) were used. Library preparation was performed using the NEBNext Ultra DNA Library Prep Kit (New England Biolabs) according to the manufacturer's protocol. Samples were mixed in equal molar ratios and sequenced on an Illumina sequencing platform.

## ChIP-seq analysis

Single-end reads were aligned to the hg19 genome using Bowtie2 (version 2.1.0). Only uniquely aligned reads were kept. Binary Alignment Map files of aligned reads were further processed using the deepTools3 (Ramirez et al, 2014). Input files were subtracted from the treatment files using the bamCompare tool, applying the signal extraction scaling method for normalization of signal to noise. Resulting signals were normalized to a mean 1× coverage to produce signal (bigWig) files. Peaks were called using the MACS 1.4 tool using default parameters.

## Differential gene expression and gene set enrichment analyses

mRNA gene expression analyses was performed within R2: genomics analysis and visualization platform (http://r2.amc.nl). We used publicly available gene expression omnibus (GEO) datasets: Neuroblastoma custom/AG44 (GEO: gse49710, n = 498; platform: ag44kcwolf), Neuroblastoma RPM/SEQC (GEO: gse62564, n = 498; platform: seqcnb1), Neuroblastoma Versteeg (GEO: gse16476, n = 88), Neuroblastoma Hiyama (GEO: gse13136; n = 30), and Normal Peripheral Glial Cells (GEO: gse99933; n = 376 [E13.5], n = 384 [E12.5]). Differentially expressed genes between clinically-relevant dichotomies in primary neuroblastoma (gse49710) were obtained by GEO2R tool (http://www.ncbi.nlm.nih.gov/geo/geo2r). Gene correlation graphs, "correlated with a single gene" data, "Parametric Gene set Enrichment" data were extracted from the R2 database. Venn diagrams were generated with the GeneVenn an BioVenn web applications (Hulsen et al, 2008). Gene set enrichment was analysed with GSEA (Subramanian et al, 2005).

## Microarray analysis

Gene expression profiles from SK-N-BE(2)c treated with vehicle and RA were generated as two one-color replicates using the whole-genome oligonucleotide microarray platform from Agilent (Agilent Technologies) as previously described (Oberthuer et al, 2006; Westermann et al, 2008). Raw microarray data were normalized using quantile normalization. Expression ratios are given as the mean of two replicates.

### RNA-seq procedure

RNA was depleted from ribosomal RNAs using the Ribo-Zero rRNA Removal Kit (Illumina) according to the manufacturer's protocol. RNA libraries were prepared using the NEBNext Ultra Directional RNA Library Prep Kit for Illumina (New England Biolabs) according to the manufacturer's protocol with the following changes: RNA was fragmented for 20 min at 94°C followed by first strand cDNA synthesis for 10 min at 25°C, 50 min at 42°C, and 15 min at 70°C. Size selection of adapter-ligated DNA was performed with a bead:DNA ratio of 0.4 (AMPure XP beads; Beckman Coulter) removing index primer and short fragments. Quality, quantity, and sizing (~320 bp) of the RNA library were analysed using a DNA High Sensitivity DNA chip run on a 2100 Bioanalyzer (Agilent Technologies). Libraries were sequenced (50 bases single-end) on the Illumina sequencing platform. The normalized gene expression values were grouped into controls and intervention samples. Fold-change was calculated both by means and trimeans ($n = 3$) and followed by a $\log_2$ transformation. Statistical significance was investigated by the rank-based Kruskal––Wallis test (https://www.jstor.org/stable/2280779?seq=1#page_scan_tab_contents) under no assumption of the underlying distribution. No filtering of data was applied based on fold-change or *P*-value cutoffs because of the low sample sizes available in the experiment.

### RNA-seq gene profile

Additional tumor RNA-seq data subset ($n = 27$) from previous NB study (Henrich et al, 2016) was applied to perform *KALRN* exon expression analysis. The reads were aligned to hg19 reference using STAR 2.5.2 tool (Dobin et al, 2013). Further per exon gene counts were computed using adjusted annotation gencode v19 and differential expression analysis along with figure generation was performed using DEXseq R package (Anders et al, 2012).

## Data Availability

All sequencing data from this publication were deposited to the European Genome-phenome Archive (Lappalainen et al, 2015), the identifier EGAS00001005023.

## Supplementary Information

## Acknowledgements

We thank Elisa Wecht, Young-Gui Park, Jochen Kreth, Steffen Bannert, Erika Kuchen, Ines Gräßer and Fanny de Vloed for technical support. This work was supported by the e:Med initiative (SYSMED-NB, grant no. 01ZX1307D to F Westermann) the German Cancer Consortium (DKTK) Joint Funding program, the ERACoSysMed grant INFER-NB to F Westermann, the German Cancer Research Center (DKFZ) intramural program for interaction projects (NCT3.0 ENHANCE to F Westermann) and the DKFZ-Heidelberg Center for Personalized Oncology (HIPO) & National Center for Tumor Diseases (NCT) Precision Oncology Program (F Westermann).

## Author Contributions

EA Afanasyeva: conceptualization, data curation, software, formal analysis, investigation, visualization, methodology, and writing—original draft, review, and editing.
M Gartlgruber: conceptualization, data curation, and investigation.
T Ryl: investigation and writing—original draft.
B Decaesteker: investigation and visualization.
G Denecker: investigation, visualization, and writing—original draft.
G Mönke: formal analysis.
UH Toprak: data curation and formal analysis.
A Florez: investigation.
A Torkov: data curation and investigation.
D Dreidax: conceptualization and data curation.
C Herrmann: data curation and formal analysis.
K Okonechnikov: data curation and formal analysis.
S Ek: investigation.
AK Sharma: formal analysis.
V Sagulenko: investigation.
F Speleman: conceptualization, data curation, and supervision.
K-O Henrich: data curation and writing—original draft.
F Westermann: conceptualization, resources, data curation, and supervision.

## Conflict of Interest Statement

The authors declare that they have no conflict of interest.

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
