## [Reviewer comments · Life Science Alliance]

Life Science Alliance

Kalirin-RAC controls nucleokinetic migration in ADRN-type neuroblastoma

Elena Afanasyeva, Moritz Gartlgruber, Tatsiana Ryl, Bieke Decaestecker, Geertrui Denecker, Gregor Mönke, Umut Toprak, Andres Florez, Alica Torkov, Daniel Dreidax, Carl Herrmann, Konstantin Okonechnikov, Sara Ek, Ashwini Sharma, Vitaliya Sagulenko, Frank Speleman, Kai-Oliver Henrich, and Frank Westermann

DOI: <https://doi.org/10.26508/lsa.201900332>

Corresponding author(s): Elena Afanasyeva, DKFZ and Frank Westermann, German Cancer Research Center

Review Timeline:

Submission Date:	2019-02-04
Editorial Decision:	2019-02-22
Revision Received:	2020-11-24
Editorial Decision:	2020-12-10
Revision Received:	2021-02-12
Accepted:	2021-02-17

Scientific Editor: Shachi Bhatt

Transaction Report:

February 22, 2019

Re: Life Science Alliance manuscript #LSA-2019-00332-T

Dr. Elena Afanasyeva
DKFZ
Im Neuenheimer Feld
280
Heidelberg 69120
Germany

Dear Dr. Afanasyeva,

Thank you for submitting your manuscript entitled "Kalirin-RAC controls nucleokinetic migration in ADRN-type neuroblastoma" to Life Science Alliance. The manuscript was assessed by expert reviewers, whose comments are appended to this letter.

As you will see, while reviewer #1 supports publication pending inclusion of a better discussion, reviewer #2 points out that the various players proposed to act in a pathway need to get better linked to each other. The reviewer also thinks that quantified data need to get provided. Given this input, we would like to invite you to submit a revised version to us, addressing all points raised by the reviewers. Please note that we would need strong support from reviewer #2 on such a revised version to allow publication here.

The typical timeframe for revisions is three months. Please note that papers are generally considered through only one revision cycle.

Thank you for this interesting contribution to Life Science Alliance. We are looking forward to receiving your revised manuscript.

Sincerely,

Andrea Leibfried, PhD
Executive Editor
Life Science Alliance
Meyershofstr. 1
69117 Heidelberg, Germany
t +49 6221 8891 502
e a.leibfried@life-science-alliance.org
www.life-science-alliance.org

B. MANUSCRIPT ORGANIZATION AND FORMATTING:

Reviewer #1 (Comments to the Authors (Required)):

Here the authors provide evidence for nucleokinetic migration in ADRN- type neuroblastoma cells. This is somehow expected since neuroblast migration require nucleokinetic migration during development. The authors beautiful demonstrate how this process require ROCK and RAC1 and

provide evidence that NUC is controlled by SOX11 and DCX.

minor comments:

the authors state in the introduction:

"understanding the mechanisms implicated in migration of DCX- positive NB can shed light on initial steps of the metastatic process in NB"

It is not clear from this paper however, if the metastatic nature of neuroblastoma originate from MES type or ADRN type, and thus if nucleokinetic migration is part of the metastatic program in high stage NB. Long distance migration of neural crest and schwann cell precursors (SCP) during embryonic development do not depend on NUC and DCX. It would be helpful to comment on this in the discussion.

Reviewer #2 (Comments to the Authors (Required)):

The manuscript by Afanasyeva et al describes morphological and molecular similarities between neuroblastoma cell lines with a sympathetic noradrenergic identity (ADRN-type NBs) and migrating neurons in the developing brain. In normal developing brains, migrating neurons extend the leading and trailing processes. During migration, the centrosome moves forward and cytoplasmic dilation is formed at the proximal region of the leading process. Subsequently, the nucleus shows elongated morphology and moves into the cytoplasmic dilation. Thus, the migrating neurons exhibit "saltatory movement". In the submitted manuscript, the authors showed that the ADRN-type NBs exhibited saltatory movement (Fig. 1F) with the formation of leading process- and cytoplasmic dilation-like structures (Fig. 1I). The authors also observed the nuclear elongation in the ADRN-type NBs, although they did not provide quantitative data. In addition, the migration of the ADRN-type NBs was shown to require Dcx, Lis1 and Rac1, both of which were previously reported to regulate neuronal migration in the developing brains. Inhibition of Rac1 and its activator, kalirin, disturbed centrosomal positioning (Fig. 6E).

These findings are interesting and informative to researchers in not only cancer biology but also developmental neuroscience. However, this reviewer finds several weak points in this manuscript. This study identified many molecules involved in the migration of ADRN-type NBs, but the epistasis of these molecules is unclear. Both the kalirin-Rac1 pathway and Sox11 upregulate the expression of Dcx, but suppression of Rac1, Sox11 and Dcx exhibits different phenotypes. The relationship between Rac1, Dcx and Sox11 in the migration of the ADRN-type NBs should be clarified. Second, several molecules, including Dcx, have previously been reported to be associated with neuroblastoma, which reduces the novelty of this manuscript. Third, the authors may confuse nucleokinesis in postmitotic neurons with interkinetic nuclear migration in neural progenitors, as described below.

Overall, this reviewer finds this manuscript potentially interesting, but many additional experiments are required to resolve the above-mentioned problems.

[Major points]

1) Knockdown of Sox11 dramatically reduced Dcx mRNA in IMR-32 cells, indicating that Sox11 is a major upstream regulator of Dcx. However, the morphological abnormalities of Dcx-knockdown cells seem to be more severe than those of Sox11-knockdown in Fig. 3H. I wonder if some of the phenotypes of the Dcx-knockdown cells might result from off-targeting effects. Does re-expression of Dcx in the Dcx-knockdown cells restore all phenotypes? In addition, the authors should also examine whether expression of Dcx could rescue the phenotypes of the Sox11- or kalirin-knockdown, which would resolve the above-mentioned first issue.

2) In line with my previous comment, the molecular relationship between Sox11 and kalirin-Rac1 is unclear, both of which act as an upstream regulator of Dcx. Do Sox11 and Rac1 independently regulate Dcx expression? Or is Sox11 an upstream (or downstream) regulator of Rac1? The authors should examine whether suppression of Sox11 (or kalirin/Rac1) affects the activity of Rac1 (or Sox11).

3) As described above, underlying mechanisms of the migration of postmitotic neurons and the interkinetic nuclear migration (IKNM) are somewhat different. While the nuclear movement in postmitotic neurons essentially requires dynein motor activity, the IKNM during G1 phase of neural progenitors depends on kinesin and myosin (and increased nuclear density at the apical region of the ventricular zone also provides the force for IKNM during G1 phase) (Genes Cells (2013) Vol.18, 176-194). This may not match the authors' conclusion, although I think there is no need that all migratory behaviors of the ADRN-type NBs and developing postmitotic neurons are similar. In addition, previous reports indicate that suppression of Rac1 activity decreases the distance between the centrosomes and nuclei (Cell Rep (2012) Vol.2, 640-651), whereas Lis1 heterozygous deficiency and Dcx knockdown increase the distance (J Cell Biol (2004) Vol. 165, 709-721. J Neurosci (2008) Vol.28, 13008-13013). It is also inconsistent with the case of the ADRN-type NBs, where Rac1 positively controls Dcx. How do you explain these incompatible results?

4) Regarding Fig. 6E, the analyzing method may not be a standard in the field of developmental neuroscience. The authors should measure the distance between the centrosomes and nuclei.

5) The immunoblot data in Fig. 5E are not good. Quantitative data are required. In addition, as I mentioned above, the nuclear elongation should also be quantified. In my eyes, the nuclei in the NBs treated with a ROCK inhibitor show abnormally elongated morphologies.

[Minor points]

6) Colocalization of kalirin with gamma-tubulin or golgi apparatus is not clear in Fig. S4B. High magnification (and high resolution) images are required.

7) I could not find "Pugh et al., 2012" and "Nishimura et al., 2017" in the reference list. Are these "Pugh et al. (2012) Nature, Vol.488, 106-110" and "Nishimura et al. (2017) Brain Sci, Vol.7, 87"?

8) The fact that Rac1 regulates the centrosomal positioning in migrating neurons has already been reported (Yang et al. (2012) Cell Rep, Vol.2, 640-651). The authors should mention this.

Item	Number of panel	Change
Fig. 1	A-D (E)(F) H (I)(J) K L (new)	----- The data from NC-CC distances are now obsolete as NUC events are presented in Fig. 1G (data from SK-N-BE(2)c_NM added) Live imaging of pseudo-3-D-assayed IMR-32_NM cells Data from DCX - and LIS1 -KD cells (from Fig. 2 (C)(D)) Data from IMR-5/75 and IMR-32 expressing G1 sensor γ-tubulin dynamics during migration in IMR-32
Fig. 2	A (previous Fig. 2E) B (C)(D)(E) F-K	The data from colcemid/cytochalasin are placed in the supplementary material (SM) Nuclear plotting from DCX - and LIS1 -KD IMR-32_NM Violin plots showing the nucleus-to-centrosome in LIS1 -KD and DCX -KD cells From Fig.3(H)(I)(J) (DCX - and LIS1 -KD IMR-32_NM) Analysis of RNA-seq data from DCX- and LIS1-KD cells in the context of primary NB
Fig. 3	(B)(C) (previous) E G H I J K (M)(N)	DCX/SOX11 expression data from primary tumours are left; DCX mRNA expression analysis in cancer cell lines, box plot with SOX11 mRNA expression, and SOX11 mRNA expression analysis from primary tumours and cancer lines are placed in the SM WB for DCX in SOX11 -KD cells Venn diagram showing IMR-32 DCX-KD^{DOWN} \cap SOX11-KD^{DOWN} overlap Representative pictures of cell morphology in DCX -KD IMR-32 were replaced From Fig. 3(I)(J) Data on the morphological asymmetry in SOX11-KD IMR-32 Additional segmentation data from SOX11-KD IMR-32 + cell velocity distribution Analysis of RNA-seq data from SOX11-KD cells in the context of primary NB The data from SH-EP_ SOX11 -TAT are placed in the SM
Fig. 4	A (J)(K)	Plots showing RHOA - and RAC1 related gene sets in DCX -KD cells RNA-seq data from the DCX-KD and RAC1-inhibitor-

		treated cells in the context of primary NB
Fig. 5	E	RAC1 pulldown in IMR-32 after SOX11 - and KALRN -KD Quantitative data for RAC1 pulldown
Fig. 6	----	----
Fig.7	new	Meta-analysis of RNA-seq data from the NUC-deficient cells in the context of primary NB
Fig. S1	C	Dcx and Lis1 expression in t-SNE-resolved E12.5 and E13.5 sympathetic precursors (Furlan et al., 2017).
Fig. S2	new	Meta-analysis of RNA-seq data from the DCX and LIS1-KD cells in the context of primary NB
Fig. S3	based on the previous version Fig. S2 (A)(B)(C) D E F (G)(H) (I)(J) K	Previous version of Fig.3(A)(B)(C) and Fig.S1(H)(I) Sox11 expression in t-SNE-resolved E12.5 and E13.5 sympathetic precursors (Furlan et al., 2017). GSEA plots showing enrichment of SOX11 and ELAVL targets in SOX11-KD IMR-32 Analysis of RNA-seq data from SOX11-KD IMR-32 in the context of primary NB The data from SH-EP_SOX11-TAT are placed in the SM
Fig. S4	Based on the previous version of Fig. S2. I	Analysis of RNA-seq data from RAC1-inhibited IMR-32 in the context of primary NB
Fig. S5	Based on the previous version of Fig. S3.	-----
Fig. S6	Based on the previous version of Fig. S4. B -> C D	Higher magnification (X63) image of kalirin-STYV, Golgi and gamma-tubulin are provided WB for TIAM1, kalirin and TRIO in SOX11-KD cells
Fig. S7	Based on the previous version of Fig. S5 C D	RAC1 pulldown replicates Kalrn expression in t-SNE-resolved E12.5 and E13.5 sympathetic precursors (Furlan et al., 2017).

	E	Sequencing of kalirin-9 3'-UTR isoform in IMR-32
Fig. S8	Based on the previous version of Fig. S6 D G H	Nucleus-to-centrosome distance in RAC1-inhibited/kalirin-suppressed/SOX11-KD IMR-32 WB for DCX in the RAC1-inhibited/kalirin-suppressed cells Data from DCX-RFP rescue in SOX11-KD , DCX-KD and KALRN-KD IMR-32
Fig. S9	previous version of Fig. S7	-----
Fig. S10		Supporting information for Fig. 7
Fig. S11		Supporting information for Fig. 7
Table S1		RNA-seq data from DCX-KD and LIS1-KD
Table S2		RNA-seq data from SOX11-KD IMR-32
Table S3		RNA-seq data from RAC1/kalirin-GEF1-suppressed IMR-32
Table S4		Analysis of DEGs from the NUC-suppressed IMR-32
Movie 1		Without JPEG compression
Movie 2		Higher resolution
Movie 3	S3 -> S3 S4	Without JPEG compression Split into two video files
Movie 4	New 4	Without JPEG compression
Movie 5	S5 -> S6 S7	Without JPEG compression Split into two video files
Movie 6		Without JPEG compression
Movie 7		Without JPEG compression
Movie 8	IMR-32 treated with TP53 siRNA	Without JPEG compression

Item	Change
SH-EP-SOX11-TAT	SH-EP_SOX11-TAT
kalirin inhibitor	kalirin-GEF1 inhibitor kalirin-GEF1 inhibition etc....

Reviewer #1 (Comments to the Authors (Required)):

Here the authors provide evidence for nucleokinetic migration in ADRN- type neuroblastoma cells. This is somehow expected since neuroblast migration require nucleokinetic migration during development. The authors beautiful demonstrate how this process require ROCK and RAC1 and provide evidence that NUC is controlled by SOX11 and DCX.

minor comments:

the authors state in the introduction: "understanding the mechanisms implicated in migration of DCX- positive NB can shed light on initial steps of the metastatic process in NB"

It is not clear from this paper however, if the metastatic nature of neuroblastoma originate from MES type or ADRN type, and thus if nucleokinetic migration is part of the metastatic program in high stage NB. Long distance migration of neural crest and Schwann cell precursors (SCP) during embryonic development do not depend on NUC and DCX. It would be helpful to comment on this in the discussion.

Response to Reviewer#1: We would like to thank the reviewer for the positive evaluation. We have extended the discussion in the manuscript in response to the comment concerning a link between NUC and metastasis in ADRN NB. Changes in the initial version of the manuscript are either highlighted for added sentences or striked through for deleted sentences, serving as the revised version.

In our manuscript, we have focused on whether NUC is involved in the migration of ADRN NB in vitro. ADRN- and MES-type NB cells appear have equal oncogenicity levels, which has been recently epitomized in the work of van Groningen and colleagues (van Groningen et al., 2019). So far, there has been no evidence proving that MES-type cells can be reprogrammed towards ADRN lineage. Yet, several works have showed that ADRN-type cells can be reprogrammed towards an MES state via MES SE-associated TFs or CRC TFs (Boeva et al., 2017; van Groningen et al., 2017; van Groningen et al., 2019) or through the downregulation of ADRN factors (Decaestecker et al., in preparation). It is still not clear whether MES and ADRN NB cells represent two uncoupled "compartments" that generate metastatic flows independently. Our data from *SOX11*-KD IMR-32 show that the reduction of motility in *SOX11*-KD IMR-32 correlates with mesenchymalisation (**Table S2, the manuscript**). This transition may result in the appearance of MES stationary (cancer) stem cells (Brabletz., 2012), which should be resolved in future studies aiming to estimate the metastatic burden in a mesenchymalised ADRN-based NB model. To accentuate the whole point with respect to

the context and direction of our manuscript, we also resolved the transcriptome in *DCX*-KD IMR-32, which revealed that mesenchymal signature did not emerge in *DCX*-KD ADRN cells, as compared to *SOX11*-KD (Table S2, the manuscript).

As the roots of MES and ADRN NB begin to be mapped out, particularly after scRNA-seq data from mouse adrenomedullary and sympathoblast cells have appeared (Furlan et al., 2017), we can also answer questions about NUC's origin in NB. Schwann cell precursor expression signatures (but not sympathoblasts, chromaffin cells or bridge cells) show high MES scores (Gartlgruber et al., in press), suggesting that MES NB originates from Schwann cell precursors. Remarkably, adult glia-derived ITSCs (Hauser et al., 2012) are also grouped with MES NB (Gartlgruber et al., in press). On the other hand, ADRN NB expression signatures show an affinity towards sympathoblasts (Gartlgruber et al., in press). Accordingly, *Dcx* is highly expressed by sympathoblasts in E12.5 and E13.5. A lower amount of *Dcx* mRNA is present in chromaffin cells and bridge population (see figure below).

E12.5

E13.5

Distribution of Dcx expression across sympathoblasts , chromaffin cells , bridge cells , and Schwann cell precursors , as retrieved from http://pklab.med.harvard.edu/cgi-bin/R/rook/nc.SS2_16_250-2/index.html and http://pklab.med.harvard.edu/cgi-bin/R/rook/nc.SS2_16_249-2/index.html (Furlan et al., 2017).

Given the presence of Dcx in sympathoblasts and chromaffin cells, we can speculate that NUC is active in these cells and might be inherited by ADRN NB. We can approach this issue by revisiting the lissencephaly and tubulinopathy cases for defects in the peripheral nervous system. Only a few cases have been reported where patients with lissencephaly also had a neurocristopathy, or Hirschsprung disease (Hikita et al., 2014; Mittal et al., 2014). One case related to a mutation in an aristaless-related gene (ARX) was reported with regard to adrenal gland hypoplasia (Bonneau et al., 2014). There is evidence that *Lis1* haploinsufficiency adversely affects the migration of (para)sympathetic preganglionic neurons, but it is not clear whether this defect also pertains to neural crest-derived neurons. Hereditary dysautonomias constitute a special type of neurocristopathy that can be mechanistically linked to NB migration. Familial dysautonomia (FD), which in most cases is caused by mutations in *IKBKAP*, was initially thought to be the result of defective neuronal migration (Naumanen et al., 2008). However, it was later suggested that neuron loss in FD was not a consequence of neuron migration failure (Jackson et al., 2014). All in all, it is unlikely that NUC is active during the development of sympathetic ganglia. Evolutionary younger than other neural crest derivatives postganglionic sympathetic elements emerge in gnathostoma through *phox2*, *ascl1* and *hand* coalescence into an expression module (Häming et al., 2011). Therefore, the correspondence between NUC migration in NB and CNS might manifest during the development of the neuronal subtypes expressing NB-like transcription factor code, i.e., hindbrain (nor)adrenergic neuronal formations (Zeisel et al., 2018). Particularly, facial branchiomotor neurons demonstrate N-C inversions when migrating tangentially (Distel et al., 2010). Also, motor neurons migrate tangentially in a *reelin*-negative region (Rossel et al., 2005), which is an interesting notion when put in the context of *RELN* negativity of advanced NB (Becker et al., 2012). Probably, migrating ADRN NB resembles motor neurons, but “gets stuck” at the stage of tangential migration-like centrosomal repolarisation. Remarkably, murine orthologue of up-regulated in *kalirin*-GEF1-inhibited cells *ETV1* is expressed in the nuclei of mouse cranial nerves (Zeisel et al., 2018). More precisely, *Etv1* expression appears in facial motor neurons during the final post-tangential stage of their

migration, the sub-nuclear segregation, and is indispensable for finalising neuronal differentiation (Zhu, Guthrie, 2013; Tenney et al., 2019). Kalirin-GEF1 inhibition upregulates low-risk-specific transcriptomic traits in ADRN NB (**Table S3, the manuscript**), which can be interpreted as a sign of a differentiation-like process. Yet, it is the kalirin paralog, Trio, that is involved in facial motor sub-nuclear segregation in mice (Backer et al., 2007). On the other hand, in xenopus, *kalrn* expression is present in the cranial nerves, while *trio* is expressed in migrating neural crest cells (Kratzer et al., 2019). In fact, kalirin and trio originated through the duplication of a proto-trio/kalirin gene in the ancestral invertebrate, but became first stabilised and functionally diversified in cyclostomata (Kratzer et al., 2019). Probably, kalirin-dependent NUC migration in ADRN NB recapitulates the relict hindbrain migration. It would thus be interesting to assess neuronal migration in the hindbrain of *kalrn* morphants. *Dcx* is present in sympathoblasts when they still undergo divisions. It is unlikely, however, that it plays a role in sympathoblast proliferation. As our data show, DCX has no significant effect on cell proliferation also in NB. DCX and LIS1 functions are not confined to NUC due to their involvement in MT transport and regeneration in PNS (Nawabi et al., 2015; Hines et al., 2018). The defects in LP and lack of significant alterations in N-C distance, which we observed in *DCX-KD* cells, conform with this idea (**Figure 2B, the manuscript**). On the other hand, as the data from the sympathetic neurons of mice show, *Dcx* function is attributed to the growth cone dynamics (Tint et al., 2009). ADRN NB terminals are reminiscent of pheochromocytoma' *varicones* (Mingorance-LeMeur et al., 2009), which are akin to exocytotically active postsynapses. So far, nothing is known about varicone propensity to transduce pro-survival and cell death signals to nucleus. On the other hand, the transcriptome of *DCX-KD* cells showed massive MT dysfunction, which might lead collapse of pro-survival retrograde signals and manifest in downregulation of catabolic signatures (**Table S1, the manuscript**). The latter highlights our future research direction, towards searching for drug-related vulnerabilities in migration-deficient NB (i.e., synthetic lethal vulnerabilities).

We checked migration in JoMa1.3 and ITSCs, which revealed mesenchymal migratory pattern in both of these cell lines (see figure below). These findings were consistent with the results of ITSC microarray expression profiling (Hauser et al., 2012; GEO: GSE30596), which showed no *DCX* expression gain in ITSCs, as compared to the *DCX*-non-expressing cell line, HUES-6 (Belzile et al., 2014). Likewise, JoMa1.3 that had been established from migratory trunk

neural crest cells (Maurer et al., 2007) migrated in a way that agrees with the absence of NUC machinery.

IMR-32 (based on Fig.1G in the manuscript)

SH-EP (based on Fig.1G in the manuscript)

JoMa1.3

ITSC_p4

Mapping of NUC events (top) in concatenated tracks from IMR-32 (n of frames = 863), SH-EP (n = 900), JoMa1.3 (n = 493) and ITSC_P4 (n = 520) and the expression of the marker of migrating neural crest cells, SOX10, was

determined by WB in nuclear protein lysates from JoMa1.3 and total protein lysates from ITSCs. For positive control, total protein lysates from melanoma cell line, MeWo (Alver et al., 2016), were used.

References

Alver TN, Lavelle TJ, Longva AS, Øy GF, Hovig E, Bøe SL. MITF depletion elevates expression levels of ERBB3 receptor and its cognate ligand NRG1-beta in melanoma. *Oncotarget*. 2016; 7:55128-55140.

Backer S., Hidalgo-Sánchez M., Offner N., Portales-Casamar E., Debant A., Fort P., Gauthier-Rouvière C., Bloch-Gallego E. Trio controls the mature organization of neuronal clusters in the hindbrain. *J Neurosci*. 2007; 27:10323-10332.

Bahi-Buisson N., Souville I., Fourniol F. J., Toussaint A., Moores, C. A., Houdusse A., Yves Lemaitre J., Poirier K., Khalaf-Nazzal R., Hully M. et al. New insights into genotype-phenotype correlations for the doublecortin-related lissencephaly spectrum. *Brain*. 2013; 136: 223–244

Becker J., Fröhlich J., Perske C., Pavlakovic H., Wilting J. Reelin signalling in neuroblastoma: migratory switch in metastatic stages. *Int J Oncol*. 2012; 41:681-689.

Belzile JP, Stark TJ, Yeo GW, Spector DH. Human cytomegalovirus infection of human embryonic stem cell-derived primitive neural stem cells is restricted at several steps but leads to the persistence of viral DNA. *J Virol*. 2014; 88:4021-4039.

Boeva V., Louis-Brennetot C., Peltier A., Durand S., Pierre-Eugène C., Raynal V., Etchevers H.C., Thomas S., Lermine A., Daudigeos-Dubus E. et al. Heterogeneity of NB cell identity defined by transcriptional circuitries. *Nat Genet*. 2017; 49: 1408-1413.

Bonneau D, Toutain A, Laquerrière A, et al. X-linked lissencephaly with absent corpus callosum and ambiguous genitalia (XLAG): clinical, magnetic resonance imaging, and neuropathological findings. *Ann Neurol*. 2002;51(3):340–349.

Brabletz T. EMT and MET in metastasis: where are the cancer stem cells? 2012 22:699-701.

Distel M, Hocking JC, Volkmann K, Köster RW. The centrosome neither persistently leads migration nor determines the site of axonogenesis in migrating neurons in vivo. *J Cell Biol*. 2010; 191:875-890.

Furlan A, Dyachuk V, Kastri ME, Calvo-Enrique L, Abdo H, Hadjab S, Chontorotzea T, Akkuratova N, Usoskin D, Kamenev D et al. Multipotent peripheral glial cells generate neuroendocrine cells of the adrenal medulla. *Science*. 2017; 357(6346).

Häming D, Simões-Costa M, Uy B, Valencia J, Sauka-Spengler T, Bronner-Fraser M. Expression of sympathetic nervous system genes in lamprey suggests their recruitment for specification of a new vertebrate feature. *PLoS ONE* 2011; 6:e26543.

Hauser S, Widera D, Qunneis F, Müller J, Zander C, Greiner J, Strauss C, Lüningschrör P, Heimann P, Schwarze H et al. Isolation of novel multipotent neural crest-derived stem cells from adult human inferior turbinate. *Stem Cells Dev*. 2012; 21: 742–756.

Hikita N, Hattori H, Kato M, Sakuma S, Morotomi Y, Ishida H, Seto T, Tanaka K, Shimono T, Shintaku H, Tokuhara D. A case of *TUBA1A* mutation presenting with lissencephaly and Hirschsprung disease. *Brain Dev*. 2014 36:159-62.

Hines, T. J., Gao, X., Sahu, S., Lange, M. M., Turner, J. R., Twiss, J. L., Smith, D. S. An essential postdevelopmental role for Lis1 in mice. *eNeuro*. 2018; doi:10.1523/ENEURO.0350-17.2018.

Jackson MZ, Gruner KA, Qin C, Tourtellotte WG. A neuron autonomous role for the familial dysautonomia gene ELP1 in sympathetic and sensory target tissue innervation. *Development*. 2014; 141: 2452-2461.

Jackson MZ, Gruner KA, Qin C, Tourtellotte WG. A neuron autonomous role for the familial dysautonomia gene ELP1 in sympathetic and sensory target tissue innervation. *Development*. 2014; 141: 2452-2461.

Jean DC, Baas PW, Black MM. A novel role for doublecortin and doublecortin-like kinase in regulating growth cone microtubules. *Hum Mol Genet*. 2012; 21: 5511–5527.

Kratzer MC, England L, Apel D, Hassel M, Borchers A. Evolution of the Rho guanine nucleotide exchange factors Kalirin and Trio and their gene expression in *Xenopus* development. *Gene Expr Patterns*. 2019; 32:18-27.

Maurer J, Fuchs S, Jäger R, Kurz B, Sommer L, Schorle H. Establishment and controlled differentiation of neural crest stem cell lines using conditional transgenesis. *Differentiation*. 2007; 75:580-591.

Mingorance-Le Meur A, O'Connor TP. Neurite consolidation is an active process requiring constant repression of protrusive activity. *EMBO J*. 2009; 28:248-260.

Mittal A, Sehgal R, Gupta R, Sharma S, Aggarwal KC. Rare association of lissencephaly with hirschprung's disease. *Astrocyte*. 2014; 1: 244-245.

Naumanen T, Johansen LD, Coffey ET, Kallunki T. Loss-of-function of IKAP/ELP1: could neuronal migration defect underlie familial dysautonomia? *Cell Adh Migr*. 2008 2: 236–239.

Nawabi H, Belin S, Cartoni R, Williams PR, Wang C, Latremolière A, Wang X, Zhu J, Taub DG, Fu X, et al. Doublecortin-like kinases promote neuronal survival and induce growth cone reformation via distinct mechanisms. *Neuron*. 2015; 88:704-719.

Rossel M., Loulier K., Feuillet C., Alonso S., Carroll P. Reelin signaling is necessary for a specific step in the migration of hindbrain efferent neurons. *Development* 2005; 132:1175–1185.

Tenney AP, Livet J, Belton T, Prochazkova M, Pearson EM, Whitman MC, Kulkarni AB, Engle EC, Henderson CE. Etv1 controls the establishment of non-overlapping motor innervation of neighboring facial muscles during development. *Cell Rep*. 2019; 29:437-452.

Tint I, Jean D, Baas PW, Black MM. Doublecortin associates with microtubules preferentially in regions of the axon displaying actin-rich protrusive structures. *J Neurosci*. 2009; 29: 10995-11010.

van Groningen T, Akogul N, Westerhout EM, Chan A, Hasselt NE, Zwijnenburg DA, Broekmans M, Stroeken P, Haneveld F, Hooijer GJ et al. A NOTCH feed-forward loop drives reprogramming from adrenergic to mesenchymal state in neuroblastoma. *Nat Commun*. 2019; 10: 1530. doi: 10.1038/s41467-019-09470-w.

Van Groningen T., Koster J., Valentijn L.J., Zwijnenburg D.A., Akogul N., Hasselt N.E., Broekmans M., Haneveld F., Nowakowska N.E., Bras J. et al. Neuroblastoma is composed of two super-enhancer-associated differentiation states. *Nat Genet*. 2017; 49: 1261-1266.

Zeisel A, Hochgerner H, Lönnerberg P, Johnsson A, Memic F, van der Zwan J, Häring M, Braun E, Borm LE, La Manno G, et al. Molecular architecture of the mouse nervous system. *Cell*. 2018; 174:999-1014.

Zhu Y, Guthrie S. Expression of the ETS transcription factor ER81 in the developing chick and mouse hindbrain. *Dev Dyn*. 2002; 225: 365–368.

Sincerely,

Elena A. Afanasyeva

Moritz Gartlgruber

Tatsiana Ryl

Gregor Mönke

Bieke Decaesteker

Geertrui Denecker

Alica Torkov

Daniel Dreidax

Carl Herrmann

Umut Toprak

Konstantin Okonechnikov

Sara Ek

Ashwini Sharma

Vitaliya Sagulenko

Frank Speleman

Kai-Oliver Henrich

Frank Westermann

Reviewer #2 (Comments to the Authors (Required)):

The manuscript by Afanasyeva et al describes morphological and molecular similarities between neuroblastoma cell lines with a sympathetic noradrenergic identity (ADRN-type NBs) and migrating neurons in the developing brain. In normal developing brains, migrating neurons extend the leading and trailing processes. During migration, the centrosome moves forward and cytoplasmic dilation is formed at the proximal region of the leading process. Subsequently, the nucleus shows elongated morphology and moves into the cytoplasmic dilation. Thus, the migrating neurons exhibit "saltatory movement". In the submitted manuscript, the authors showed that the ADRN-type NBs exhibited saltatory movement (Fig. 1F) with the formation of leading process- and cytoplasmic dilation-like structures (Fig. 1I). The authors also observed the nuclear elongation in the ADRN-type NBs, although they did not provide quantitative data. In addition, the migration of the ADRN-type NBs was shown to require *Dcx*, *Lis1* and *Rac1*, both of which were previously reported to regulate neuronal migration in the developing brains. Inhibition of *Rac1* and its activator, kalirin, disturbed centrosomal positioning (Fig. 6E).

These findings are interesting and informative to researchers in not only cancer biology but also developmental neuroscience. However, this reviewer finds several weak points in this manuscript. This study identified many molecules involved in the migration of ADRN-type NBs, but the epistasis of these molecules is unclear. Both the kalirin-*Rac1* pathway and *Sox11* upregulate the expression of *Dcx*, but suppression of *Rac1*, *Sox11* and *Dcx* exhibits different phenotypes. The relationship between *Rac1*, *Dcx* and *Sox11* in the migration of the ADRN-type NBs should be clarified. Second, several molecules, including *Dcx*, have previously been reported to be associated with neuroblastoma, which reduces the novelty of this manuscript. Third, the authors may confuse nucleokinesis in postmitotic neurons with interkinetic nuclear migration in neural progenitors, as described below.

Overall, this reviewer finds this manuscript potentially interesting, but many additional experiments are required to resolve the above-mentioned problems.

Thank you very much for your thoughtful and thorough review of our manuscript. We have read the comments and concerns very carefully. The idea that *DCX* is an NB-relevant gene has indeed been highlighted in several correlative studies. We believe that our work is the first attempt to explore whether there is a functional link between the presence of *DCX* and the migration mode in NB. Apart from technical issues, it appears to us that the reviewer's concerns focus on two important issues: 1) the link between *SOX11*, *DCX* and kalirin; and 2) the transitions between MES and ADRN states in KD and drug-treated cells. Below, we respond in detail to the reviewer's comments and describe our additional experiments. Changes in the initial version of the manuscript are either highlighted for added sentences or striked through for deleted sentences in the revised version.

[Major points]

1) Knockdown of Sox11 dramatically reduced Dcx mRNA in IMR-32 cells, indicating that Sox11 is a major upstream regulator of Dcx. (*) However, the morphological abnormalities of Dcx-knockdown cells seem to be more severe than those of Sox11-knockdown in Fig. 3H. (**) I wonder if some of the phenotypes of the Dcx-knockdown cells might result from off-targeting effects. Does re-expression of Dcx in the Dcx-knockdown cells restore all phenotypes? (***) In addition, the authors should also examine whether expression of Dcx could rescue the phenotypes of the Sox11- or kalirin-knockdown, which would resolve the above-mentioned first issue.

These are two excellent points, particularly the one concerning the DCX rescue.

(*) The phenotype in *DCX*-KD IMR-32 is not “more severe” but distinctly different in nature. As we wrote in the previous version of our manuscript, GO term and motif analysis of *SOX11*-KD IMR-32 up-regulome revealed enrichment for AP-1 and ETS1 targets as well as EMT hallmark (Table S2, the manuscript and Figure R1), suggesting NUC(ADRN)-to-MES transition. This transition was also observed in *SOX11*-KD IMR-32 using an independent siRNA (Decaesteker et al., publication in preparation). In line with this observation, *SOX11*-KD facilitated the inhibition of migration; however, evidence was found concerning slow, NUC-independent migration mode (Figures 3K and 3L) that was accompanied with the depletion of tubulin-related expression signatures and upregulation of actin-related signatures (Table S2).

Figure R1. Gene set enrichment analysis (GSEA) in *SOX11*-KD vs control cells showing enrichment of EMT-related signatures (q-values < 0,05)

ADRN-to-MES (or, in the context of cell migration, NUC-to-MES) transitions have been already considered as unfavorable situations with regard to tumor evolution (Boeva et al., 2017; van Groningen et al., 2017). Based on the cell phenotype and the migration mode, we proposed that *DCX*-KD IMR-32 did not undergo NUC-to-MES transition. To verify this finding, we performed RNA-seq profiling in *DCX*-KD IMR-32 using *LIS1* siRNA as an additional control. *DCX*-KD drove alterations in expression of several NB TFs in IMR-32. Particularly, moderate

upregulation of two ADRN factors, *SOX11* and *GATA3*, and MES factors *SOX9*, *ETS1*, *NFIA* and *GLI3* (Boeva et al., 2017; van Groningen et al., 2017) was observed. Yet, genes containing binding sites for MES NB factors were not overrepresented in the upregulated genes (**Table S1**). Consistently, no evidence for MES program was found in the up-regulomes of *DCX*- (and *LIS1*-KD) cells (**Table S1, Figures S2A (the manuscript) and R2**). This confirmed our idea about the maintenance of neuroblastic program in *DCX*-KD cells.

DCX-KD IMR-32 *LIS1*-KD IMR-32 (q-values: n.s.)

Figure R2. GSEA in *DCX*-KD and *LIS1*-KD vs control IMR-32 showing EMT-related signatures.

Nevertheless, the downregulated genes in the *DCX*-KD cells were enriched for several expression hallmarks, suggesting that *DCX* regulated the transcription. This was further supported by overrepresentation of genes with bindings motifs for the major transcription factors, MYC, TP53 and E2F1, in the downregulated genes (**Table S1**). The overrepresentation of MYC targets also indicated possible functional overlap between *DCX* and *SOX11*. This point has been mentioned in the revised version of our manuscript. Yet, we are puzzled as to how *DCX* regulates transcription in NB. It is known that *Dcx* regulates motor-driven neuronal transport mediated by Kif1a (Liu et al., 2012), thus having the potential to control neurite-to-nucleus signaling and hence transcription.

We would like to mention that statistical analysis shows that *DCX* expressions defined by RNA-seq in stage 4S and stage 4 belong to different distributions according to KS test (Figure R3). The microarray data show a similar trend, but *p*-values are higher than 0.05 (Figure R3; Fig. S3B, manuscript). *DCX* expression in primary NB might be influenced by an “yet-unknown factor”. For example, high *DCX* mRNA dispersions can stem from a lineage specific-expression of *DCX* in primary NB. Therefore, the reduced levels of *DCX* mRNA in stage 4S NB is not a passenger event. We would like to keep these data in the revision letter.

RNAseq; GEO: gse62564

Microarray; GEO: gse49710

Figure R3. Boxplot demonstrating *DCX* expression in primary NB (top, GEO: gse62564): *p*-values: (4,MYCNnonamp;4S): 0.0004; (MNA;4S): 7.338e-15; two-sample Kolmogorov-Smirnov test.

(bottom, GEO: gse49710). *p*-values: (4,MYCNnonamp;4S): 0.13; (MNA;4S): 0.13; two-sample Kolmogorov-Smirnov test.

(**) we took into consideration that the off-target effects of *DCX* RNAi have been noted previously (albeit in the experiments with a shRNA and not siRNA; Baek et al., 2014). Following the reviewer's suggestion, we complemented our video analysis of *DCX*-KD by inserting two additional siRNAs[§] to *DCX* (Figure R4), which revealed a migration-defective phenotype (Figure R5). We would like to keep these data within this letter instead of the main text.

Figure R4. WB for DCX and accumulated migration distances in IMR-32_NM cells in controls and after 48h of *DCX* RNAi (Ambion siRNAs:145587 (#1) and 145588 (#2) (16 hr). Mean values + SD are presented.

control siRNA

DCX siRNA#2

Figure R5. Phase contrast images of IMR-32 after 48h of *DCX* RNAi (Ambion siRNAs:145587 (#2)

§ Additional *DCX* siRNAs (145587 and 145588) were purchased from Thermo Fisher Scientific.

(***) *DCX* rescue experiments indeed make sense, as our RNA-seq in *SOX11*-KD IMR-32 suggests that *DCX* is one of the major NUC-relevant *SOX11* targets. We approached this question by expressing RFP-tagged *DCX* (Tanaka et al., 2004; **Figure R6**) in IMR-32_NM and subjecting the derived cells to video analysis, which revealed a lack of a statistically significant compensation of migration defect in *SOX11*-KD cells (**Figure S8H**). The most plausible explanation for this situation is the MES nature of *SOX11*-KD, which makes *DCX* irrelevant (“lineage-alienated”) to the cell migration. Supporting this explanation, the rescue effect was clearly observed in the *DCX*-KD cells (with a siRNA targeting 5’-UTR of *DCX*). We then asked whether ectopic *DCX* acquired cell cycle-related functions in the *SOX11*-KD cells. Cell tracking

showed that expression of RFP-tagged DCX did not affect cell cycle duration in control and *SOX11*-KD IMR-32 cells (data not shown).

Figure R6. DCX-RFP localisation in interphase (top) and mitotic (bottom) IMR-32 cells

2) In line with my previous comment, the molecular relationship between Sox11 and kalirin-Rac1 is unclear, both of which act as an upstream regulator of Dcx. Do Sox11 and Rac1 independently regulate Dcx expression? Or is Sox11 is an upstream (or downstream) regulator of Rac1? The authors should examine whether suppression of Sox11 (or kalirin/Rac1) affects the activity of Rac1 (or Sox11).

This is a very interesting suggestion. Our RNA-seq data showed that several RAC1-GEF-encoding genes, including *TIAM1* and *TRIO*, are potentially regulated by *SOX11* (p -values by Kruskal-Wallis t-test: **KALRN, 0.13; TIAM1, 0.049; TRIO, 0.049; Table S2**). We checked this finding using WB, which revealed kalirin and *TIAM1* downregulation in IMR32 upon *SOX11*-KD (**Figure S6D**). We did not observe the induction of previously annotated *TRIO* isoforms upon *SOX11*-KD. We also checked *SOX11* expression in RAC1-/kalirin-GEF1-inhibited and *KALRN*-KD IMR32 cells and did not observe any significant changes in the *SOX11* level. Next, we searched for *SOX11* targets in our RNA-seq data from RAC1/kalirin-GEF1-inhibited cells (**Table S3**). The first attempt to characterize NB-relevant *SOX11* targets has been already undertaken

(Decaesteker et al., under revision). We discuss SOX11 targets and SOX11-interacting proteins in the context of previously published works (Kuo et al., 2015; Heim Birgit, 2014 etc). One of the SOX11 interactors, Kleefstra-syndrome-associated EHMT1 (**p-values: kalirin-GEF1 inh#2, 0.025; kalirin-GEF1 inh#1, 0.17; RAC1 inh, 0.09**) (Heim Birgit, 2014), was found within the list of downregulated genes. *EHMT1* and *SOX11* are co-expressed in primary NBs (R = 0.410; **Figure R7**). Other potential candidates downregulated in RAC1/kalirin-GEF1-inhibited cells included *SOX11*-co-expressed, a transcriptional repressor-encoding *RCCD1* (R= 0.311, **p-values: kalirin-GEF1 inh#1, 0.025; kalirin-GEF1 inh#2, 0.14; RAC1 inh, 0.1**), as well as *KIF6* (**p-values: RAC1 inh, 0.05; kalirin-GEF1 inh#2, 0.0526; kalirin-GEF1 inh#1, 0.0525**).

Figure R7. Scatter plot showing the correlation between the expressions of *SOX11* and *EHMT1* (498 patients; GEO: gse49710).

Altogether, these data support the idea that *SOX11* regulates *DCX* and the genes encoding RAC1-GEFs, however, we could not exclude a negative feedback in case of the *SOX11-DCX* link. We also think that other RAC1-/kalirin-regulated candidates could be found among *SOX11* protein interactors.

We also checked whether forced expression of the *DCX* construct corrected migration defects induced by *KALRN-KD*, which revealed the compensation of migration upon *DCX-RFP* transfection (**Figure S8H**).

3) As described above, underlying mechanisms of the migration of postmitotic neurons and the interkinetic nuclear migration (IKNM) are somewhat different. While the nuclear movement in postmitotic neurons essentially requires dynein motor activity, the IKNM during G1 phase of neural progenitors depends on kinesin and myosin (and increased nuclear

density at the apical region of the ventricular zone also provides the force for IKNM during G1 phase) (Genes Cells (2013) Vol.18, 176-194). This may not match the authors' conclusion, although I think there is no need that all migratory behaviors of the ADRN-type NBs and developing postmitotic neurons are similar.

Likewise, we are intrigued by the idea of IKNM activity in NB. As we state in our manuscript (“IMR5-75 expressing a FUCCI cell cycle sensor [Ryl et al., 2017] and growing asynchronously, which revealed tendency for migration in the G1 phase [Figure 2K; the manuscript.]”), the fact that NB cells are less migratory in the S/G2 phase along with DCX involvement is an indirect evidence of neuron-like migration, rather than IKNM. Indeed, this does not necessarily mean that nuclear migrations do not take place during S and G2 phases. It is known that interkinetic nuclear migrations are minimal during S-phase (Hayes, Nowakowski; 2000). To resolve this question, we used IMR-32 expressing G1 marker (Figure R8) and mapped the nuclear and cytoplasmic centers. We did not observe a drastic switch in NNC/NCC mapping after the cells passed through G1 (Figure R9), which suggested similar migration mechanisms (Figure 2K).

Figure R8. Boxplot showing cell migration in 1528 (G1 phase) and 985 (S/G2 phases) sequential timepoints (5 min per timepoint, respectively) from tracings of 22 cells from IMR-32 expressing the G1 cell cycle sensor; *p*-value by two-sample Kolmogorov-Smirnov test: 2.2e-16.

Figure R9. An exemplary track from an expressing the G1 cell cycle sensor IMR-32 cell that going through G1/S transition.

Dcx controls basally directed nuclear movement in rat brain progenitor cells (Carabalona et al., 2016). While IKNM is not amenable to examination in dissociated cultures due to the absence of adherens junctions present *in vivo* (LaMonica et al., 2013), differential knockdown

(KD) of *DCX*, controlling G1-specific, kinesin-dependent NUC (Carabalona et al., 2016), and of *LIS1*, controlling G2-specific, dynein-dependent NUC and spindle assembly during IKNM (Tsai et al., 2005; Yingling et al., 2008; Carabalona et al., 2016), helped to tell IKNM from NUC in NB.

To consolidate all the data, we checked the effect of *CDK5* RNAi on the migration in IMR-32. Apparently, *CDK5* plays no role during IKNM, but becomes indispensable during post-mitotic neuronal migration (Chae et al., 1997; Ohshima et al., 1996; Nishimura et al., 2014). We checked the effect of *CDK5* RNAi on the migration in IMR-32_NM spheroids, which showed inhibition of migration (Figure R10). *CDK5*-KD cells had NUC defects (Figure R11, top; compare to the controls in the manuscript), but generation of cell projections was intact (Figure R11, bottom; compare to the controls in the manuscript). *CDK5*-KD retained IMR-32-like NNC/NCC angle frequency distribution (Figure R12). The correlation between cell velocity and NUC footprint was retained. In general, the migration defects were less severe than in the *DCX*-KD cells.

Figure R10. Random walk plots (left) and accumulated migration distances (middle) of control and *CDK5*-KD IMR-32_NM cells (48 hr of RNAi + 6 hr of tracking, 5-min intervals). Mean migration distances and SD are presented. Welch t-test *p*-value: 0.04. Western blot analysis of *CDK5* levels in IMR-32 after *CDK5* RNAi (right).

CDK5 siRNA was purchased from Santa Cruz. *CDK5* antibody was purchased from Cell Signaling.

Figure R11. Representative [NC-CC] plots (top) in IMR-32_NM after RNAi against *SOX11*. Mapping of positive and negative noise-corrected NC-CC distances, 0–40° and 140–180° signatures (two or more sequential frames within the same angle block) (bottom) from concatenated tracks from *CDK5-KD* IMR-32_NM cells.

Figure R12. NNC/NCC angle frequency distribution in *CDK5-KD* IMR-32_NM cells (left). Correlation plots (right) between cell velocity and NUC footprint (weighted mean NUC distance) in *CDK5-KD* IMR-32_NM cells.

Further, expression of a construct encoding γ -tubulin fused to mCherry in IMR-32 showed that nuclei surpassed γ -tubulin signals during migration (**Figure 1L**), which was in agreement with the neuronal nucleus-centrosome (N-C) inversion mechanism (Umeshima et al., 2007), but not IKNM.

Thus, two programs, the nucleokinesis and the cell cycle, merge in ADRN NB. We would like to keep these data in the revision letter. Yet, these experiments do not provide an answer to the question whether G2 phase-specific mechanisms of nuclear movement are active in NB (particularly, a nuclear pore-mediated mechanism that involves RANBP2-BICD2 (Baffet et al., 2015)). We found that *RANBP2* and *BICD2* were co-expressed in NB ($R = 0.648$; Figure R13). A number of missense mutations and the presence of CpG methylation (as documented in the CCLE database) in *RANBP2* gene in NB indicated that CDK1-BICD2-RANBP2 functionality might not be intact in NB.

Figure R13. Box plots demonstrating *RANBP2* and *BICD2* expression in primary NB: (stages 1, 2, 3, 4 and 4S) and scatter plot showing the correlation between the expressions of *RANBP2* and *BICD2* (498 patients; GEO: gse49710).

Missense mutation: LAN-6, TGW, NB10, NB17, KPNYS, NBTU110

CpG methylation: SIMA, SKNBE2, SKNDZ, CHP126, SKNFI, KPNYN, NH6, NB1, CHP212, KPNSI9S, KPNRTBM1

List of NB cell lines with *RANBP2* missense mutations and methylation (as taken from Cancer Cell line Encyclopedia (CCLE)).

The second issue is the centrosome motility observed in NB cells, which might provide a clue to the whole problem. During the G1 phase of IKNM, centrosome is tethered at the apical surface via a primary cilium, which persists during the cell cycle until late G2 (Spear and Erickson, 2012). On the other hand, in migrating post-mitotic cortical neurons, primary cilia are highly dynamic (Higginbotham et al., 2012). This is a very interesting topic that should be pursued in further studies. In this respect, it is well worth mentioning that a ciliary dysfunction disorder, Bardet-Biedl syndrome, manifests in neural crest migration defects and is associated with Hirschsprung's disease (Tobin et al., 2008). Moreover, one of the primary regulators of cilia disassembly, Aurora A (Korobeynikov et al., 2017), is deregulated in NB (Faisal et al., 2011).

Overall, the question is also related to NUC origin in NB. **A detailed response to this question is provided in the revised discussion.**

In addition, previous reports indicate that suppression of Rac1 activity decreases the distance between the centrosomes and nuclei (Cell Rep (2012) Vol.2, 640-651), whereas Lis1 heterozygous deficiency and Dcx knockdown increase the distance (J Cell Biol (2004) Vol. 165, 709-721. J Neurosci (2008) Vol.28, 13008-13013). It is also inconsistent with the case of the ADRN-type NBs, where Rac1 positively controls Dcx. How do you explain these incompatible results?

During the manuscript preparation, we checked the low-resolution images of the *LIS1*-KD cells and did not find extreme centrosome "overshoots" (Tanaka et al., 2004), which already seemed a discrepancy with regard to the previous publications. Following the reviewer's suggestion, we measured the nucleus-to-centrosome distance in *DCX*-, *LIS1*-, *SOX11*-KD and RAC1-/kalirin-suppressed IMR-32 cells at a higher magnification (**Figure 2B and S8D**). This revealed that RAC1- or kalirin-suppression indeed reduced the nucleus-to-centrosome distance (**Figures R14 and R15; Figure S8D**). Also, we observed a higher variability in the nucleus-to-centrosome distance in *LIS1*-KD, but not in *DCX*-KD IMR-32 cells. This conforms with the results of video analysis which demonstrated cell rounding and faint projections in *DCX*-KD IMR-32. We also noticed that fewer centrosomes were present distally in the cells with elongated nuclei after *DCX*-KD (see figure below). This is somewhat common (although coupled also with nuclear rounding) in RAC1- and kalirin-suppressed as well as *SOX11*-KD

IMR-32 cells (Figure 6E), which led us to conclude that both DCX and RAC1/kalirin might regulate the centrosome translocation in NB cells.

Figure R14. Violin plots demonstrating the nucleus-to-centrosome distance in control, *DCX*-KD and *LIS1*-KD IMR-32 cells (*p*-values: *DCX* siRNA: n.s. (n = 168), *LIS1* siRNA; 0.009898 (n = 168); by Welch t-test).

Figure R15. Violin plots demonstrating the nucleus-to-centrosome distance in control, *RAC1*-, kalirin-GEF1-suppressed and *SOX11*-KD IMR-32 cells. (*p*-values: *RAC1* inh: 0.0001028 (n = 90), kalirin-GEF1 inhibitor#1: 1.933e-06 (n = 90), kalirin-GEF1 inhibitor#2: 1.191e-05 (n = 180), *KALRN* siRNA: 0.0001527 (n = 114), *SOX11* siRNA; 2.388e-05 (n = 71); *p*-values by Welch t-test).

4) Regarding Fig. 6E, the analyzing method may not be a standard in the field of developmental neuroscience. The authors should measure the distance between the centrosomes and nuclei.

We have added the data from centrosome-to-nucleus measurements (Figure 2B and S8D) but would like to retain Figure 6E. This is because we think that this panel provides useful information, and thus we respectfully ask the reviewer to re-consider his/her point.

5) The immunoblot data in Fig. 5E are not good. Quantitative data are required. In addition, as I mentioned above, the nuclear elongation should also be quantified. In my eyes, the nuclei in the NBs treated with a ROCK inhibitor show abnormally elongated morphologies.

Thank you very much for the opportunity to complement our RAC1 pulldown data with additional replicates and the results from *SOX11*-KD (Figure 5E; Figure S7C). Given the upregulation of RAC1 neighborhood in *SOX11*-KD IMR-32 (Table S2), we weren't sure about the outcome of pulldown experiments in *SOX11*-KD. We think the results captured the re-distribution of RAC1 activity in the *SOX11*-KD cells. The data on nuclear elongation are provided as the distribution of nuclear roundness across the cell population (several hundred cells) in Figure 6G. This panel shows that RAC1/kalirin-GEF1 inhibition increases nuclear roundness.

Yes, we also noticed that ROCK inhibition affects nuclear shape, albeit in a different way (Figure R16). We noticed that nuclei look groove-less in the cells after ROCK inhibition, while the grooves are still visible in the nuclei of RAC1-/kalirin-GEF1-suppressed cells (Figure R17). Also, ROCK-treated nuclei may look elongated on the kymographs (Figure 4G)). This is due to image scaling along the x-axis.

Figure R16. DAPI staining showing changes in the nucleus shape in IMR-32 after ROCK inhibition.

IMR-32; vehicle-treated cells

IMR-32; ROCK-inhibitor-treated cells

Figure R17. 3-D rendering of nuclei in control and ROCK inhibitor-treated cells.

Figure R18. Boxplots demonstrating nuclear roundness in IMR-32 cells treated with RAC1-, kalirin-GEF1 inhibitors or ROCK inhibitor. Data represent three independent experiments (819, 1008, 375, 772 and 1023 cells, respectively).

As **Figure R18** demonstrates there is no nuclear elongation in ROCK inhibitor-treated cells. As ROCK is not the major focus of our paper, we have included these data in the response letter but not the main text.

Minor points

6) Colocalization of kalirin with gamma-tubulin or golgi apparatus is not clear in Fig. S4B. High magnification (and high resolution) images are required.

We have addressed this issue and are happy to provide high-magnification images taken from an independent experiment (**Figure S6B**). These data confirm our previous observation that kalirin is not a centrosomal protein in NB.

7) I could not find "Pugh et al., 2012" and "Nishimura et al., 2017" in the reference list. Are these "Pugh et al. (2012) Nature, Vol.488, 106-110" and "Nishimura et al. (2017) Brain Sci, Vol.7, 87"?

Thank you, this is now corrected. "Pugh et al., 2012", corresponds to a publication by Pugh et al. named "The genetic landscape of high-risk neuroblastoma," published in Nature Genetics, along with the following work: Nishimura et al., (2017) Brain Sci, Vol.7, 87. Morphological and Molecular Basis of Cytoplasmic Dilation and Swelling in Cortical Migrating Neurons.

8) The fact that Rac1 regulates the centrosomal positioning in migrating neurons has already been reported (Yang et al. (2012) Cell Rep, Vol.2, 640-651). The authors should mention this.

We included the work of Yang et al., published in 2012 ("POSH localizes activated Rac1 to control the formation of cytoplasmic dilation of the leading process and the neuronal migration to the results in the context of RAC1"), in the context of the RAC1/kalirin part.

Below we also describe extra-experiments not related to reviewer's questions, but relevant to the revision work.

1) First of all, we highly encourage our referees to go through our analysis of NUC-deficient cells and our conclusions about epigenetic downregulation in NUC-deficient cells. We invest a lot of hopes to this direction.

2) Next, according to PCA plotting (Decaesteker et al., unpublished observation, **Figure R18**) the clone we chose for the previous manuscript ("B6") version does not cluster with the

other SH-EP_SOX11-TAT clones. We are still not sure whether this was a cloning artefact, or a result of SH-EP heterogeneity that might have something to do with the NB lineages. Finally, we decided to replace the results from clone “B6” with results from three other clones (previously Figure 3K, new Figure S3K-L). All three clones demonstrate slight but statistically significant migration increase upon induction of SOX11 expression.

Figure R18. Principal Component Analysis (PCA) plot demonstrating SH-EP_SOX11-TET clones

Figure R19. Mean fluorescence intensity in SH-EP_SOX11-TAT from control and doxycycline-induced SH-EP_SOX11-TAT (exemplary tracks from three different groups).

Figure R20. Boxplots demonstrating cell velocity ($\mu\text{m}\times\text{min}^{-1}$) in vehicle ($n = 2065$) and doxycycline-induced ($n=2112$) SH-EP_SOX11-TAT. p -value by Kruskal-Wallis test: $2.2\text{e-}16$. No noise-correction was applied.

3) At the moment, we are not sure whether *SDHA* is a competent reference gene since our RNA-seq analyses in *DCX*-KD IMR-32 revealed that this gene was co-regulated with *DCX*. We re-analysed our RT-qPCR data from the *DCX*-KD cells using *HPRT1*. Our western blots show that there was no co-regulation of *DCX* protein in *LIS1*-KD IMR-32 (**Figure R20**).

Figure R20. WB for DCX and LIS1 in IMR-32 after transfection (WB for DCX and LIS1 in IMR-32 (Trizol-isolated protein; RNA-seq replicate). LIS1 AB was from Cell Signaling.

References

- Baek ST, Kerjan G, Bielas SL, Lee JE, Fenstermaker AG, Novarino G, Gleeson JG. Off-target effect of doublecortin family shRNA on neuronal migration associated with endogenous microRNA dysregulation. *Neuron*. 2014; 82: 1255-1262.
- Baffet AD, Hu DJ, Vallee RB. Cdk1 Activates pre-mitotic nuclear envelope dynein recruitment and apical nuclear migration in neural stem cells. *Dev Cell*. 2015; 33:703-716.
- Banerjee D., Gryder B., Liu Z., Bagchi S., Chou H-C., Sindiri S., Khan J., Thiele CJ. Sequential Expression of Lineage Specific Transcription Factors Drives Differentiation in High-Risk Neuroblastoma (HR-NB). ANR 2018 San Francisco Abstract. 349.
- Carabalona A., Hu D. J., Vallee R. B. KIF1A inhibition immortalizes brain stem cells but blocks BDNF-mediated neuronal migration. *Nature Neurosci*. 2016; 19; 253-262.
- Chae T, Kwon YT, Bronson R, Dikkes P, Li E, Tsai LH. Mice lacking p35, a neuronal specific activator of Cdk5, display cortical lamination defects, seizures, and adult lethality. *Neuron*. 1997;18: 29-42.
- Dzieran J, Garcia AR, Westermark UK, Henley AB, Sánchez EE, Träger C, Johansson HJ, Lehtiö J, Arsenian-Henriksson M. Opposing roles of ER α and MYCN in neuroblastoma. *Proc Natl Acad Sci USA*. 2018; 115: E1229-E1238.
- Faisal A, Vaughan L, Bavetsias V, Sun C, Atrash B, Avery S, Jamin Y, Robinson SP, Workman P, Blagg J, Raynaud FI et al. The aurora kinase inhibitor CCT137690 downregulates MYCN and sensitizes MYCN-amplified neuroblastoma in vivo. *Mol Cancer Ther*. 2011; 10: 2115-2123.
- Furlan A, Dyachuk V, Kastrioti ME, Calvo-Enrique L, Abdo H, Hadjab S, Chontorotzea T, Akkuratova N, Usoskin D, Kamenev D et al. Multipotent peripheral glial cells generate neuroendocrine cells of the adrenal medulla. *Science*. 2017; 357: 6346.
- Hayes NL, Nowakowski RS. Exploiting the dynamics of S-phase tracers in developing brain: interkinetic nuclear migration for cells entering versus leaving the S-phase. *Dev Neurosci*. 2000; 22: 44-55.
- Heim, Birgit. (2014). SOX11 interactome analysis: Implication in transcriptional control and neurogenesis (Doctoral dissertation). Retrieved from <https://publikationen.uni-tuebingen.de/xmlui/bitstream/handle/10900/59876/Dissertation%20Birgit%20Heim.pdf>.
- Higginbotham H, Eom TY, Mariani LE, Bachleda A, Hirt J, Gukassyan V, Cusack CL, Lai C, Caspary T, Anton ES. Arl13b in primary cilia regulates the migration and placement of interneurons in the developing cerebral cortex. *Dev Cell*. 2012; 23: 925-938.
- Korobeynikov V, Deneka AY, Golemis EA. Mechanisms for nonmitotic activation of Aurora-A at cilia. *Biochem Soc Trans*. 2017; 45: 37-49.
- Kuo PY, Leshchenko VV, Fazzari MJ, Perumal D, Gellen T, He T, Iqbal J, Baumgartner-Wennerholm S, Nygren L, Zhang F, et al. High-resolution chromatin immunoprecipitation (ChIP) sequencing reveals novel binding targets and prognostic role for SOX11 in mantle cell lymphoma. *Oncogene*. 2015; 34: 1231-1240.
- LaMonica B.E., Lui J.H., Hansen D.V., Kriegstein A.R. Mitotic spindle orientation predicts outer radial glial cell generation in human neocortex. *Nat Commun*. 2013; 4: 1665. doi: 10.1038/ncomms2647.
- Liu JS, Schubert CR, Fu X, Fourniol FJ, Jaiswal JK, Houdusse A, Stultz CM, Moores CA, Walsh CA. Molecular basis for specific regulation of neuronal kinesin-3 motors by doublecortin family proteins. *Mol Cell*. 2012; 47:707-721.
- Nishimura YV, Shikanai M, Hoshino M, Ohshima T, Nabeshima Y, Mizutani K, Nagata K, Nakajima K, Kawauchi T. Cdk5 and its substrates, Dcx and p27kip1, regulate cytoplasmic dilation formation and nuclear elongation in migrating neurons. *Development*. 2014; 141:3540-3550.
- Ohshima T, J M Ward, Huh CG, Longenecker G, Veeranna, H C Pant, Brady RO, Martin LJ, Kulkarni AB. Targeted disruption of the cyclin-dependent kinase 5 gene results in abnormal corticogenesis, neuronal pathology and perinatal death. *Proc Natl Acad Sci USA*. 1996, 93: 11173-11178.

Pugh TJ, Morozova O, Attiyeh EF, Asgharzadeh S, Wei JS, Auclair D, Carter SL, Cibulskis K, Hanna M, Kiezun A, et al. The genetic landscape of high-risk neuroblastoma. *Nat Genet.* 2013; 45:279-284.

Spear PC, Erickson CA. Apical movement during interkinetic nuclear migration is a two-step process. *Dev Biol.* 2012; 370: 33-41.

Tanaka, T., Serneo, F. F., Higgins, C., Gambello, M. J., Wynshaw-Boris, A., & Gleeson, J. G. Lis1 and doublecortin function with dynein to mediate coupling of the nucleus to the centrosome in neuronal migration. *J Cell Biol.* 2004; 165: 709-721.

Tint I, Jean D, Baas PW, Black MM. Doublecortin associates with microtubules preferentially in regions of the axon displaying actin-rich protrusive structures. *J Neurosci.* 2009; 29: 10995-1010.

Tobin JL, Di Franco M, Eichers E, May-Simera H, Garcia M, Yan J, Quinlan R, Justice MJ, Hennekam RC, Briscoe J et al. Inhibition of neural crest migration underlies craniofacial dysmorphism and Hirschsprung's disease in Bardet-Biedl syndrome. *Proc Natl Acad Sci U S A.* 2008; 105: 6714-6719.

Tsai J.W., Chen Y., Kriegstein A.R., Vallee R.B. LIS1 RNA interference blocks neural stem cell division, morphogenesis, and motility at multiple stages. *J Cell Biol.* 2005; 170: 935-945.

Umeshima H., Hirano T., Kengaku M. Microtubule-based nuclear movement occurs independently of centrosome positioning in migrating neurons. *Proc Natl Acad Sci USA* 2007; 104: 16182-16187.

van Groningen T, Akogul N, Westerhout EM, Chan A, Hasselt NE, Zwijnenburg DA, Broekmans M, Stroeken P, Haneveld F et al. A NOTCH feed-forward loop drives reprogramming from adrenergic to mesenchymal state in neuroblastoma. *Nat Commun.* 2019; 10: 1530.

van Groningen T., Koster J., Valentijn L.J., Zwijnenburg D.A., Akogul N., Hasselt N.E., Broekmans M., Haneveld F., Nowakowska N.E., Bras J. et al. Neuroblastoma is composed of two super-enhancer-associated differentiation states. *Nat Genet.* 2017; 49: 1261-1266.

van Waveren C, Moraes CT. Transcriptional co-expression and co-regulation of genes coding for components of the oxidative phosphorylation system. *BMC Genomics.* 2008; 14; 9.

Yang T, Sun Y, Zhang F, Zhu Y, Shi L, Li H, Xu Z. POSH localizes activated Rac1 to control the formation of cytoplasmic dilation of the leading process and neuronal migration. *Cell Rep* 2012; 2: 640-651.

Yingling J, Youn YH, Darling D, Toyo-Oka K, Pramparo T, Hirotsune S, Wynshaw-Boris A. Neuroepithelial stem cell proliferation requires LIS1 for precise spindle orientation and symmetric division. *Cell.* 2008; 132:474-486.

Sincerely,

Elena A. Afanasyeva

Moritz Gartlgruber

Tatsiana Ryl

Gregor Mönke

Bieke Decaesteker

Geertrui Denecker

Alica Torkov

Daniel Dreidax

Carl Herrmann

Umut Toprak

Konstantin Okonechnikov

Sara Ek

Ashwini Sharma

Vitaliya Sagulenko

Frank Speleman

Kai-Oliver Henrich

Frank Westermann

December 10, 2020

RE: Life Science Alliance Manuscript #LSA-2019-00332-TR

Dr. Elena Afanasyeva
DKFZ
Im Neuenheimer Feld
280
Heidelberg 69120
Germany

Dear Dr. Afanasyeva,

Thank you for submitting your revised manuscript entitled "Kalirin-RAC controls nucleokinetic migration in ADRN-type neuroblastoma". We would be happy to publish your paper in Life Science Alliance pending final revisions necessary to meet our formatting guidelines.

Along with the points listed below, please also attend to the following:

- please check the Author contributions and make sure that the author list and their contributions match between the eJP system, the manuscript file and in the Author contribution section in the manuscript file (for eg. Dr. Flores is listed under the Au Contribution section in the ms, but is not listed as an author on eJP or in the first page of the ms file; and Drs. Sharma, Sagulenko and Speleman's contributions are not listed in the Au Contribution section in the ms file)
- please re-title the 'Competing financial interest' section as 'Conflict of Interest'
- please include a Data Availability section, under which the accession numbers for the large datasets need to be included. In the same vein, please deposit the large datasets (ChIP seq and RNA seq) in publicly available databases.
- please separate the figure legends and supplementary figure legends into 2 separate sections within the ms text file
- please add callouts for Figure S3A, S3F, S4A-D, and S4J
- please add the ORCID IDs for both corresponding authors. Both should have received the instructions for how to do so in their respective email inboxes
- please add a legend for Figure 3J
- please add titles and legends to the supplemental table files

A. FINAL FILES:

B. MANUSCRIPT ORGANIZATION AND FORMATTING:

Sincerely,

Shachi Bhatt, Ph.D.
Executive Editor
Life Science Alliance
<https://www.lsjournal.org/>
Tweet @SciBhatt @LSAJournal

Reviewer #1 (Comments to the Authors (Required)):

The authors have have carefully addressed my comments and thus and I recommend publication in the journal.

Reviewer #2 (Comments to the Authors (Required)):

The authors added a great deal of data to the manuscript. I think the revised manuscript has been improved, and now I recommend the current version of the manuscript for publication.

February 17, 2021

RE: Life Science Alliance Manuscript #LSA-2019-00332-TRR

Dr. Elena A Afanasyeva
DKFZ
Im Neuenheimer Feld
280
Heidelberg 69120
Germany

Dear Dr. Afanasyeva,

Thank you for submitting your Research Article entitled "Kalirin-RAC controls nucleokinetic migration in ADRN-type neuroblastoma". It is a pleasure to let you know that your manuscript is now accepted for publication in Life Science Alliance. Congratulations on this interesting work.

DISTRIBUTION OF MATERIALS:

Again, congratulations on a very nice paper. I hope you found the review process to be constructive and are pleased with how the manuscript was handled editorially. We look forward to future exciting submissions from your lab.

Sincerely,

Shachi Bhatt, Ph.D.

Executive Editor

Life Science Alliance

<https://www.lsjournal.org/>

Interested in an editorial career? EMBO Solutions is hiring a Scientific Editor to join the international Life Science Alliance team. Find out more here -

https://www.embo.org/documents/jobs/Vacancy_Notice_Scientific_editor_LSA.pdf